# Im2Painting: Economical Painterly Stylization

## Abstract

This paper presents a new approach to stroke optimization for image stylization based on economical paintings. Given an input image, our method generates a set of strokes to approximate the input in a variety of economical artistic styles. The term economy in this paper refers a particular range of paintings that use few brushstrokes or limited time. Inspired by this, and unlike previous methods where a painting requires a large number of brushstrokes, our method is able to paint economical painting styles with fewer brush strokes, just as a skilled artist can effectively capture the essence of a scene with relatively few brush strokes. Moreover, we show effective results using a much simpler architecture than previous gradient-based methods, avoiding the challenges of training control models. Instead, our method learns a direct non-linear mapping from an image to a collection of strokes. Perhaps surprisingly, this produces higher-precision results than previous methods, in addition to style variations that are fast to train on a single GPU.

## 1 Introduction

A skilled painter can convey the essence of a scene with a few brush strokes (Figure 2). In art, this is sometimes called *economy*: conveying a lot with just a few lines or strokes. The tension between content and media has been hypothesized as a major source of art's appeal (Pepperell, 2015), and economy heightens this tension: something that just looks both like a compelling landscape and like a splattering of brush strokes can be more intriguing than an accurate and detailed painting of the same landscape.

Painting is one of the oldest processes by which humans form high-level representations of scenes to translate their perception of the world to static images. Abstraction and (thinking through) making are fundamental components of both painting processes and certain cognitive abilities (Clark & Chalmers, 1998; Ingold, 2013; Lake et al., 2015; 2017). In this context, considerable recent effort has focused on training neural agents to optimize stroke layouts (Huang et al., 2019; Jia et al., 2019; Singh & Zheng, 2021; Schaldenbrand & Oh, 2021), and only a few of them aimed for painting abstracted representations of an input image (Ganin et al., 2018; Mellor et al., 2019).

Painterly stylization begins with an input photograph, and generates a set of brush strokes to create a stylized version of the input. Many existing painterly stylization algorithms create an appealing "impressionistic" appearance by placing many scattered paint strokes that roughly approximate the image, e.g., Haeberli (1990); Litwinowicz (1997); Hertzmann (1998); Zou et al. (2021); Liu et al. (2021). These methods are not economical; they can capture fine details only by drawing thousands of small strokes.

We want to emphasize that *economy* is not just "one style." Many common painting styles involve considerable precision in their brush strokes, e.g., consider the way the examples in Figure 1 use a few strokes that are carefully aligned to image features, curves follow contours, and so on. Existing painting algorithms are not economical; they often spray the canvas with strokes, using many strokes to reproduce the appearance of the image well, but often with an "impressionistic" look as a result; one could not imagine capturing the styles of, say, Edward Hopper or Wayne Thiebaud with these techniques. We argue that economy is a good test of how well an algorithm can optimize strokes to match images well—a prerequisite for many natural painting styles.

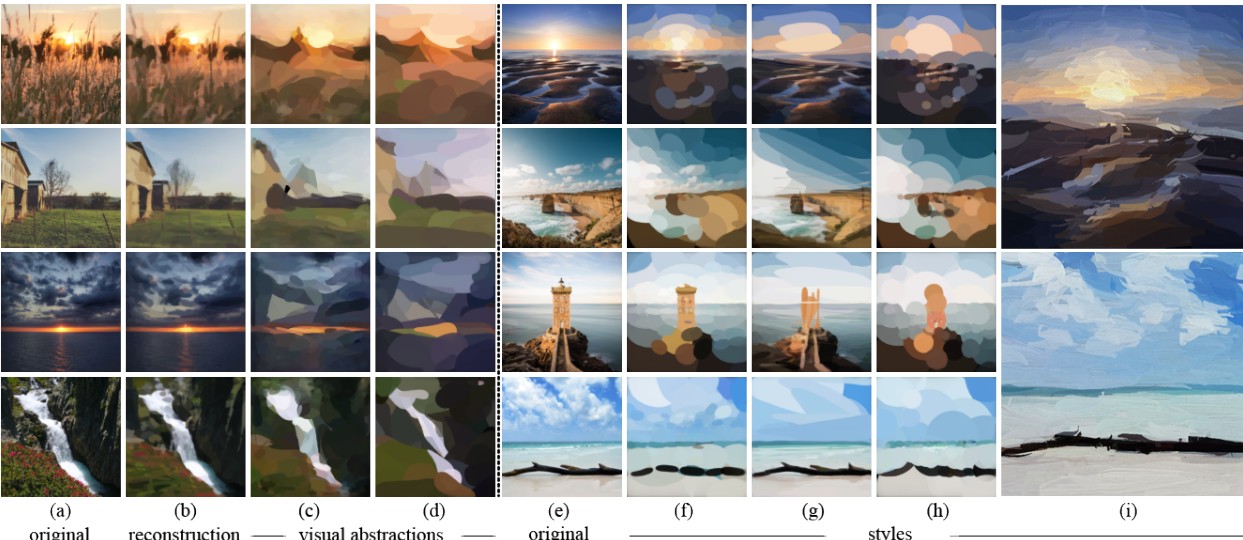

|     |     |     |     |     |     |     |     |     |
| (a) | (b) | (c) | (d) | (e) | (f) | (g) | (h) | (i) |
| original | reconstruction | —— visual abstractions —— | | original | —— | | styles | —— |

Figure 1: Painting Stylization. Given an input image (a) and (e), our method optimizes brush stroke arrangements in a variety of levels of abstraction and styles. Left: by loosening constraints from precise reconstructions (b) or reducing the number of strokes in the output painting, we can produce images with a range of abstraction levels (c,d). Right: adding loss terms and constraints can achieve different styles. (f) approximates the image by placing short strokes while capturing structure. (g) learns to distribute more strokes in the foreground, resulting in detailed foreground and coarse background. (h) places thick and short strokes resulting in a coarser, yet smooth style. (i) oil-painting texture applied using thin-plate spline algorithm. All paintings use 300 strokes, except for (d), which uses 100 strokes.

Hence, we view the problem of economical painting as foundational for many painting styles, because it is about developing general-purpose optimization techniques, and our method presents an advance on this problem. A fundamental difference with previous work is that they approximate an input image via addition of many (in the order of thousands) strokes, while our method approximates an input image via alignment of fewer strokes (no more than 300) to image features. We take inspiration from economical paintings (see Figure 2).

We demonstrate variations of artistic abstraction of the input image, i.e., very visually similar to the input in one extreme (Figure 1(b)), and abstracted representations in the other extreme (Figure 1(c-d)). We also show variations of artistic *style* by varying the techniques used for the artistic abstraction, such as by preferring shorter strokes (Figure 1(h)) or injecting noise into the strokes (Figure 6(left: e)). Varying the losses and constraints produces predictable variations in style, e.g., optimizing every stroke produces more accurate image reconstructions but less visual abstraction.

Though our method could be applied to any class of images, we focus on landscape painting. Landscapes in art history span a range from realism to abstraction, as seen in the work of painters like J. M. W. Turner, Claude Monet, and Richard Diebenkorn, each of whom created increasingly abstract landscapes as they got older. We also show quantitative comparisons on facial portraiture.

Our contributions are summarized as follows:

- We present an approach for economical-style paintings. Our method approximates stylistic versions of the input with fewer but with a more efficient placement of strokes than previous methods, where they often scatter thousand of strokes to approximate an input image.

- Our painting method uses a new and simpler architecture based on a direct non-linear mapping from a given photograph to a set of strokes, unlike previous methods that generally model an agent that takes decisions at each time step.

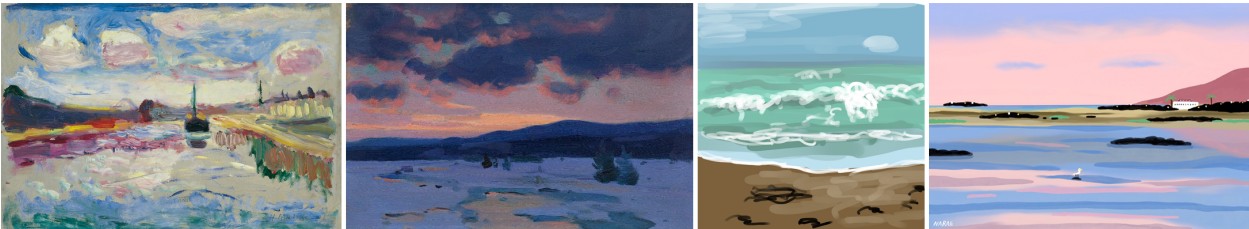

Figure 2: Economical painting styles motivate our work. These landscape paintings—oil paintings by Henri Matisse (Canal du Midi, 1899) (leftmost), Clarence Gagnon (Crépuscule D'hiver, 1915) (center-left) and two from digital painters—illustrate how an artist can convey a scene with relatively few brush strokes.

- We demonstrate a controllable framework to achieve different styles, resulting in visually appealing paintings.

## 2 Related Work

The earliest painterly stylization algorithms applied hand-designed rules to produce impressionistic effects Haeberli (1990); Litwinowicz (1997); Hertzmann (1998). The first optimization-based painting method Hertzmann (2001) produced effective stroke placements, but with cumbersome optimization heuristics. EM-like packing algorithms have been used for non-overlapping stroke primitives Hertzmann (2003); Rosin & Collomosse (2013), such as stipples Secord (2002) and tile arrangements Kim & Pellacini (2002). A variety of related methods can be used to stylize 3D models, e.g., Meier (1996); Schmid et al. (2011). Painterly stylization is a high-dimensional optimization problem, with challenging local minima, and these methods struggle to efficiently achieve economical results for overlapping strokes.

Many recent methods employ neural optimization (Zhang et al., 2018; Ganin et al., 2018; Mellor et al., 2019; Huang et al., 2019; Jia et al., 2019; Singh & Zheng, 2021; Schaldenbrand & Oh, 2021; Liu et al., 2021), in which a network is trained to produce a set of strokes from an input image. These methods optimize a loss on the output image without any supervision, i.e., no training paintings. Neural approaches use either a pre-trained neural renderer, or a non-neural differentiable renderer, and often use perceptual losses Zhang et al. (2018) rather than L2 or L1.

Reinforcement Learning algorithms train a painting agent (Ganin et al., 2018; Mellor et al., 2019; Huang et al., 2019; Jia et al., 2019; Singh & Zheng, 2021; Schaldenbrand & Oh, 2021), responsible for generating a sequence of strokes through interactions with a critic network. The learning signal comes in the form of rewards from the critic network, normally trained using adversarial learning (Goodfellow et al., 2014). Some RL methods focus on accurate depiction (Huang et al., 2019; Singh & Zheng, 2021). Huang et al. (2019) can accurately reconstruct images, but requires thousands of tiny strokes to do so, and is stylistically limited (see Figure 6(right: c,d)). Schaldenbrand & Oh (2021) show variations on this style for use by a robotic arm. LpaintB by Jia et al. (2019) is a combination of RL and self-supervised learning that produces coarser versions of the input image. Singh & Zheng (2021) also focus on accurate depiction using a semantic guidance pipeline, slightly improving image reconstructions, but also without particularly artistic results.

A few RL methods do create more artistic abstractions. SPIRAL by Ganin et al. (2018) introduced adversarially-trained actor-critic algorithms, though results were blurry. SPIRAL++ by Mellor et al. (2019) presented several improvements, achieving a very intriguing range of abstracted image styles. However, their method provides little or no interpretable control over the style and level of precision in the reconstruction, whereas we focus on precise and controllable styles.

A few recent methods have explored direct differentiable optimization without RL or neural stroke generation. DiffVG by Li et al. (2020) and Stylized Neural Painters (SNP) by Zou et al. (2021) directly optimize the stroke arrangement with gradient-based optimization. Such methods produce attractive outputs, and SNP

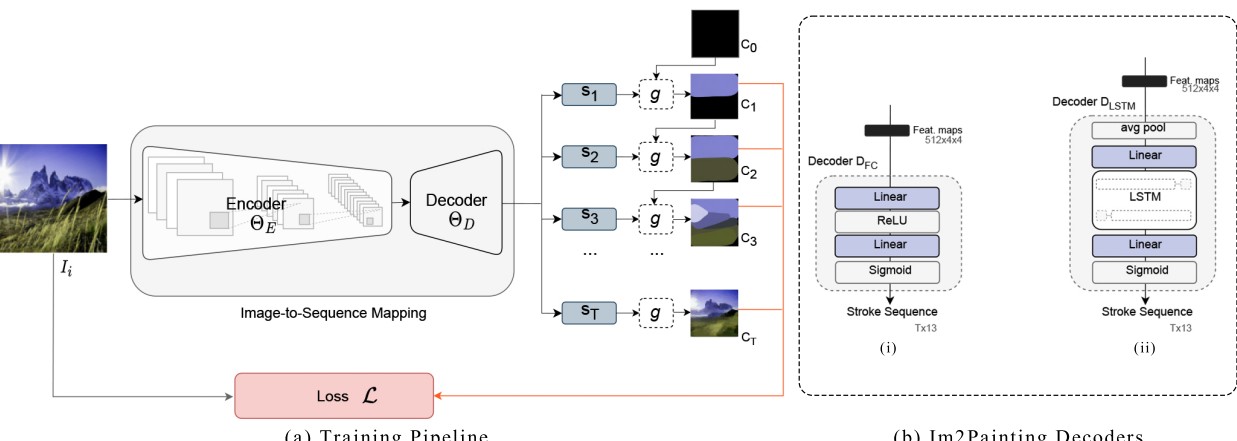

(a) Training Pipeline        (b) Im2Painting Decoders

Figure 3: Model overview. We train an end-to-end painting network without stroke supervision. (a) The framework takes an image as input and produces a stylized painting. There are two main components, an encoder function that extracts visual features from an input image, and a decoder that maps such features into a sequence of brushstroke parameters. Finally, a differentiable renderer sequentially renders the brushstroke parameters onto a canvas. An energy function $\mathcal{L}$ is an aggregation of loss functions that, in combination with the network type and optimization strategy control the style of the output. (b) Im2Painting Decoders. We propose two decoders, $D_{FC}$ for visual abstractions, consisting on 2 fully connected layers followed by non-linearities (i) and $D_{LSTM}$ which adds an LSTM for precise reconstructions and economical artistic styles (ii).

produces style variation through different stroke parameterization and textures. However, these methods generate thousands of strokes per image, and the optimization process can be very slow for each painting.

Paint Transformer by Liu et al. (2021) models the painting problem as stroke prediction, using a CNN-Transformer model without the need for training with off-the-shelf datasets, showing excellent generalization capability. Both SNP and Paint Transformer share a very similar texturized painting look to the detriment of very precise depictions. Neither of these methods address stylistic control beyond applied texture, or abstractions given by limitations of the optimization approach. RNN-based methods have also been used to train line drawing and sketching styles that are quite distinct from painting (Ha & Eck, 2017; Kingma & Welling, 2013; Zheng et al., 2019; Mo et al., 2021).

## 3   Optimization Framework

Given an input image $I \in \mathbb{R}^{3xHxW}$ and a stroke budget $T$, our method outputs a sequence of brush strokes that, when rendered sequentially onto a blank canvas $C_0$, produce a painting representation $C_T$ of $I$, as shown in Figure 3. The level of precision of the final canvas as well as the painting style depend on the loss function $\mathcal{L}$, the stroke budget, and the optimization strategy. Except where stated, we use $T = 300$ strokes as the economical stroke budget in our paper.

In contrast to previous work that either achieves good reconstructions but do not control style (Huang et al., 2019), or focuses on stylization via texture (Liu et al., 2021; Zou et al., 2021) or via brushstroke parameterization and style-transfer technique (Zou et al., 2021), our goal is to learn a network that produces economical paintings in different and controllable styles, from reconstructions to visual abstractions, by varying loss functions and constraints. That is, our main interest resides in the ability to generate artistic styles given by the concept of economy in art. We maintain the same stroke parameterization and we do not use external style images to achieve style variations.

### 3.1   Model Architecture

We propose a direct non-linear mapping network $f : \mathcal{I} \to \mathcal{S}^{T \times 13}$ from image to brush stroke sequence, in contrast to previous agent models. We split the mapping into an encoder and a decoder (Figure 3(a)). The encoder extracts feature maps using the four residual blocks of a Resnet-18 (He et al., 2015), not including the last average pooling layer, to extract a vector of feature maps $X \in \mathcal{X}^{512 \times 4 \times 4}$ from a given image $I$. The decoder $D$ transforms these feature maps into a fixed sequence of stroke parameters $s = \{s_1, s_2, ..., s_T\}$, that is, a $T \times 13$ vector. A differentiable renderer $g$ renders the stroke parameters onto a canvas, and a loss function $\mathcal{L}$ evaluates how well the painting fits the desired artistic style.

In this paper, we use two different decoder architectures, $D_{\mathrm{FC}}$ and $D_{\mathrm{LSTM}}$, as shown in Figure 3(b). The first decoder architecture, $D_{\mathrm{FC}}$, is a stack of non-linear fully-connected (FC) layers. $D_{\mathrm{FC}}$ resizes and transforms $X$ into a fixed sequence of strokes using 2 FC layers, with ReLU after the first layer and a sigmoid function after the last layer. We use $D_{\mathrm{FC}}$ for visual abstractions.

The second decoder, $D_{\mathrm{LSTM}}$, uses a bidirectional LSTM layer in combination with fully-connected layers. $D_{\mathrm{LSTM}}$ uses average pooling on $X$ to get a vector $H \in \mathbb{R}^{512}$ before feeding it into its first FC layer. The first FC layer expands $H$ into $W \in \mathbb{R}^{512 \times T}$, forming the sequence of vectors that are the input to the LSTM layer. A second FC layer followed by a sigmoid outputs the sequence of brushstroke parameters. We use $D_{\mathrm{LSTM}}$ for those styles that need to preserve geometric and semantic attributes. Ablations show that $D_{\mathrm{FC}}$ works better for visual abstractions, whereas $D_{\mathrm{LSTM}}$ is more effective at producing precise reconstructions and artistic styles (see appendix for a thorough comparison, more technical details, and visualizations about the difference between the two decoders).

### 3.2   Stroke Parameterization and Differentiable Renderer

We use the stroke parameter representation and differentiable renderer provided by Huang et al. (2019). Each stroke is parameterized by a 13-dimensional tuple that encodes start, middle and end points of a quadratic Bézier curve, radii and transparency at start and end points, and RGB color of the stroke. We use a neural renderer $g$ that has been previously trained to approximate a non-differentiable renderer. $g$ consists of four fully connected layers and six convolutional layers, and takes in the sequence of actions, one by one, and sequentially updates the initial canvas $C_0$. The rendered painting $C_T$ is then passed to a loss function $\mathcal{L}$, which generally is a combination of different losses that determine the style of the painting.

While previous methods use an $N \times N$ canvas subdivision (Huang et al., 2019; Liu et al., 2021; Zou et al., 2021) to paint small strokes on $N^2$ patches in parallel, our network operates at the canvas level at all times.

### 3.3   Loss Functions and Optimization

Training our model with different loss functions, weights and optimization schemes produces different styles. Given an input image $I$, we define the objective as a combination of different loss functions:

$$\mathcal{L}_{\mathrm{painting}} = \lambda_1 \mathcal{L}_{\mathrm{perc}} + \lambda_2 \mathcal{L}_{\mathrm{guidance}} + \lambda_3 \mathcal{L}_{\mathrm{style}} \tag{1}$$

where the first term is a perceptual loss comparing the output painting $C_T$ to the input image, the second term is a pixel loss applied to intermediate canvases, the third term is optional style losses on strokes, and the $\lambda$ values are constant weights. We explain the first two terms below, and describe the optional style losses in the next section.

**Perceptual loss.**   We use a standard form of perceptual loss (Zhang et al., 2018). Such losses produce more appealing results than pixelwise losses like $L_2$, which lead to blurry paintings. Specifically, let $V_{ij} = \{V_{ij}^1, ..., V_{ij}^k\}$ and $W_{ij} = \{W_{ij}^1, ..., W_{ij}^k\}$ be a set of $k$ feature vectors extracted from image $I$ and canvas $C_T$, respectively, we use cosine similarity as follows:

$$\mathcal{L}_{\mathrm{perc}} = -\cos\theta = \frac{1}{K} \sum_k^K \sum_{ij} V_{ij}^k W_{ij}^k \, / \|V_{ij}^k\| \, \|W_{ij}^k\| \tag{2}$$

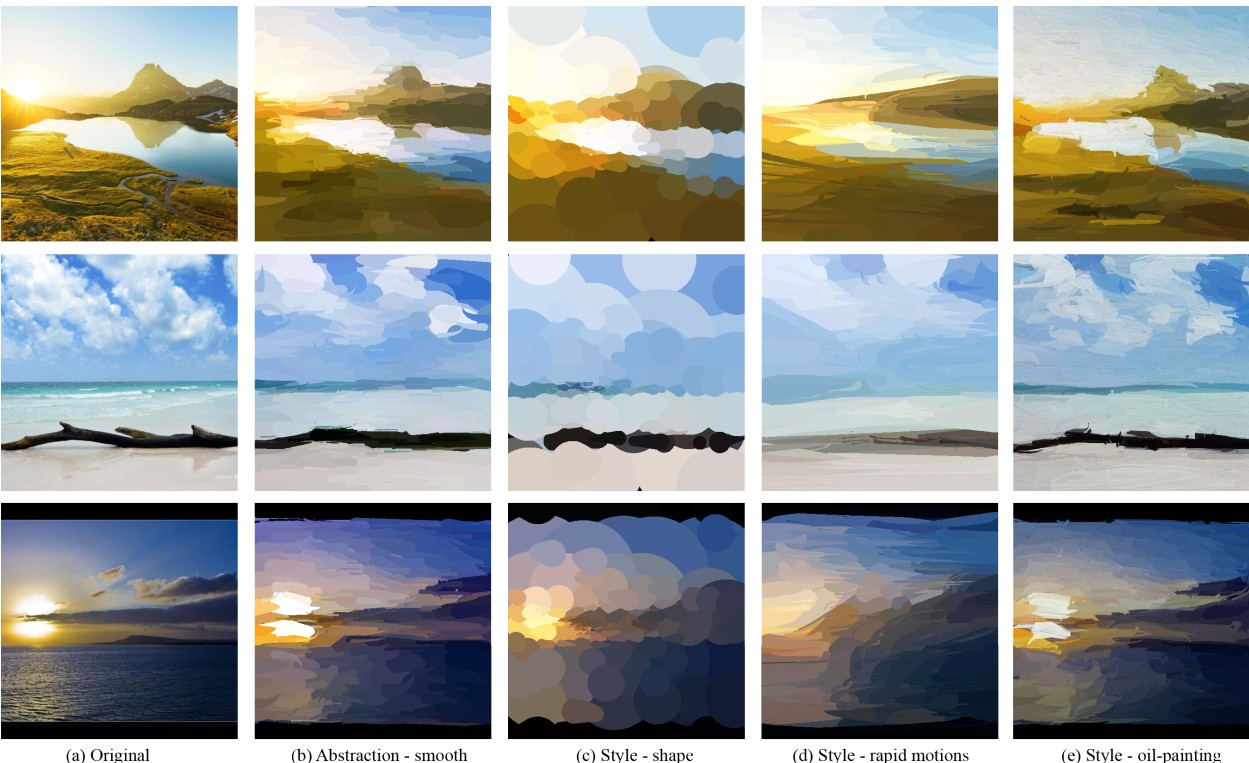

(a) Original     (b) Abstraction - smooth     (c) Style - shape     (d) Style - rapid motions     (e) Style - oil-painting

Figure 4: 512x512 300-stroke economical paintings. These examples illustrate results with higher resolution, and show how our method can provide nuanced, interpretable control over styles through varying losses and other interpretable factors. (b) shows a basic style by smooth visual abstraction. (c) adds constraints on thickness and length that reduces the number of small strokes. (d) adds random noise to strokes resulting in a scratchy style. (e) oil-painting style.

where $i, j$ index the spatial dimensions of the feature maps $V$ and $W$, and $K$ the extracted layers from VGG16 trained on ImageNet Simonyan & Zisserman (2014). Specifically, we use layers 1, 3, 6, 8, 11, 13, 15, 22 and 29. Perceptual loss generally captures high-frequency parts of the images, which are not usually represented by pixel losses.

**Guidance losses.** Previous methods typically apply losses to the output image. However, propagating losses to earlier strokes leads to slow convergence. RL methods partially bypass this with critic functions, but training the critic is itself challenging, leading to very long training times. Instead, we apply a pixelwise loss to every intermediate stage of the canvas to help "guide" the optimization. Let $C_t$ be the painting after the $t$-th stroke is added, so that $C_0$ is the initial canvas. Then,

$$\mathcal{L}_{\text{guidance}} = \sum_{t=1}^{T} \mathcal{L}_{\text{pixel}}(I, C_t) \tag{3}$$

where $\mathcal{L}_{\text{pixel}}$ computes the pixelwise $L_1$ distance between the input image and $C_t$. $L_1$ loss is enough to capture the difference in overall composition and color in image space.

## 4 Economical Painterly Styles

In this section, we show a range of artistic styles, from accurate reconstructions or realistic-looking paintings, to variations of visual abstraction given by loosening constraints of our precise network and varying

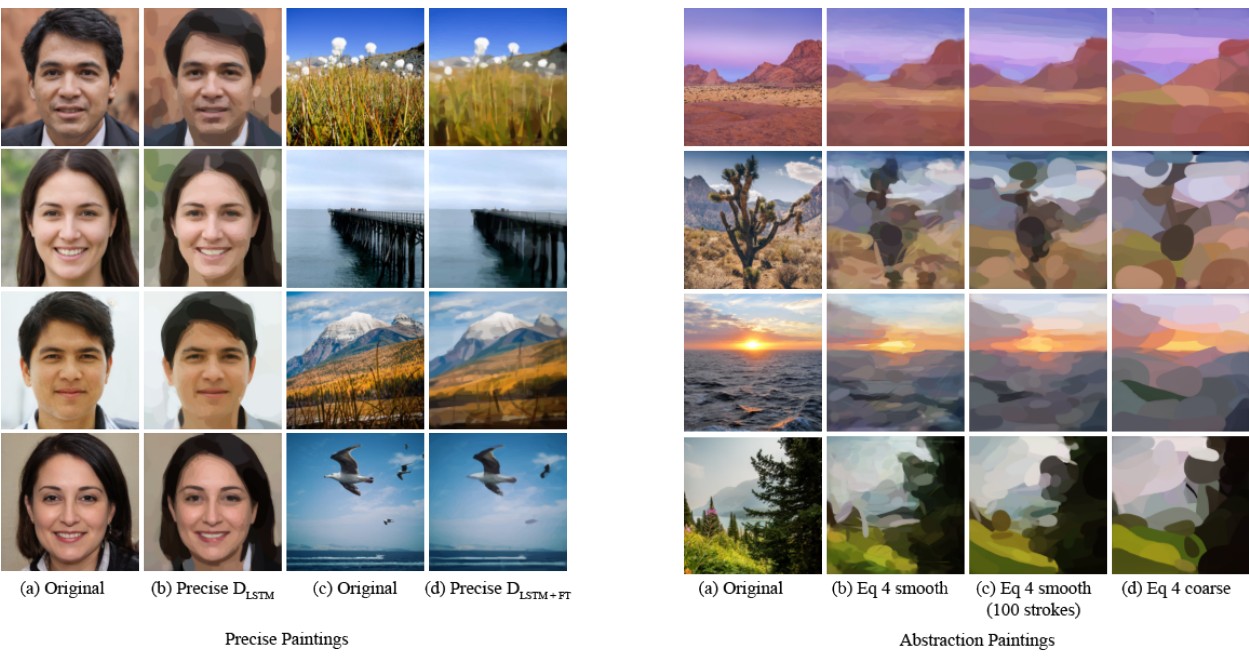

Figure 5: Precise Paintings and Visual Abstractions. (Left) Our network is able to paint accurate reconstructions with a limited budget of only 300 strokes. This is achieved using decoder $D_{LSTM}$ and following Equation (1). (Right) We can increase visual abstraction by decreasing stroke budgets and/or using coarser loss functions. (a) Input image. (b) 300 strokes using $\mathcal{L}_1$ on all intermediate canvases. (c) 100 strokes using $\mathcal{L}_1$ on all intermediate canvases. (d) 300 strokes evaluating $\mathcal{L}_1$ only on the final frame. We use $D_{FC}$ for all abstractions.

optimization guidance, stroke budget, shape constraints or rapid drawing motions. We show that a limited stroke budget of 300 strokes is enough to achieve all our styles, and we can increase visual abstraction by sparsely backpropagating gradients from fewer intermediate canvases.

**Training details.**  Different optimization approaches require different settings. For precise depiction painting, we train on 8 Nvidia Tesla V100 16-GB GPU with batch size of 168, and takes approximately 24h to train. For other optimization methods, we train on a single GPU with a batch size of 24, converging after 5h on average. We use Adam optimizer with a learning rate of 0.0002 and betas 0.5 and 0.99. We use 100000 landscapes training images gathered from Chen et al. (2018); Skorokhodov et al. (2021); Zhu et al. (2017) and 200000 CelebA Liu et al. (2015) training images. Figures 4 and 7 show examples at 512x512 of our artistic styles.

**Accurate Reconstruction.**  We first describe a basic style that shows the ability of our model to achieve precise reconstructions using an economical budget. Even though we find this realistic style less artistically interesting, we use it as a testbed to assess the validation of our method, because reconstruction is a measure of economy. To achieve this precise or realistic style, we set the loss function to $\mathcal{L}_{\text{realistic}} = \lambda_1 \mathcal{L}_{\text{perc}} + \lambda_2 \mathcal{L}_{\text{guidance}}$, where $\lambda_2 = 1$ and we set $\lambda_1$ to 0.1 on landscapes and to 0.001 on face portraits. We do not use any style-specific loss term, $\mathcal{L}_{style}$, and we use decoder $D_{\text{LSTM}}$. We show paintings on this style in Figure 1(b), Figure 5(b,d), Figure 6(right: b), and ablations in the appendix. Landscapes require a higher $\lambda$ due to a higher variance of the distribution of images. To achieve nearly-perfect reconstructions on landscapes, we add a fine-tuning step consisting on a 100-step optimization of the stroke parameters output by the network. Note that such step is only necessary for high-accurate reconstructions on landscapes, but not necessary for face images, nor any other visual style (see Figure 5(left) (b) does not use fine-tune step, (d) uses fine-tune step. Refer to appendix for more details).

**Visual Abstractions.** We can achieve stylistic variations and artistic abstraction by reducing optimality, while preserving the semantics of the input image. This is done by varying the optimization process, i.e, backpropagating gradients at different timesteps. We find that using only $L_1$ loss reduces visual fidelity, as does applying the loss only on the final canvas $C_T$ rather than on all canvases as in Equation 3. The loss we use for these experiments is $\mathcal{L}_{\text{abst}} = \lambda_2 \mathcal{L}_{\text{guidance}}$, and our guidance loss is defined as follows:

$$\mathcal{L}_{\text{guidance}} = \begin{cases} \sum_{t=1}^{T} ||I - C_t||_1 & \text{smooth} \\ \sum_{t=1}^{T/k} ||I - C_k t||_1 & \text{medium} \\ ||I - C_T||_1 & \text{coarse} \end{cases} \tag{4}$$

Where smooth, medium, and coarse define the degree of abstraction within an economical style. Likewise, we can further control the level of abstraction by varying the number of strokes used; fewer strokes produce coarser abstractions. Examples of these ranges of abstraction are shown in Figure 5(right). This figure, arranged left to right in increasing abstraction, shows how economical paintings are achieved using different constraints. We can see the effect of the number of strokes and the optimization schema in the last two columns. In (c), paintings are generated by using just 100 strokes and optimizing using a smooth guidance. However, the last column (d) uses 300 strokes within a coarse schema, which results in a more abstracted painting than the previous paintings, even using 3x more strokes. The coarse loss is computed only in the final frame, which causes strokes in early stages of the process to not contribute much to the final painting. This is further shown in appendix Figure 12. For these experiments we use $D_{\text{FC}}$. We also show the results of applying an oil painting texture on smooth abstraction using the thin plate spline algorithm (Zhao & Zhu, 2011; Barrodale et al., 1993) in Figure 1(i), Figure 4(e) and Figure 9(e), Figure 18 in Appendix.

**Stroke Styles.** In the above experiments, the network is free to paint with any stroke size. In this section, we add optional style losses to penalize stroke parameters that lie outside desired ranges, i.e., penalizing strokes that are too large. Specifically, let $S_p$ be some property of stroke shape. We set a shape threshold $T_s$ to penalize properties that are too large.

$$\mathcal{L}_{\text{shape}} = \begin{cases} 0 & \text{if } S_p < T_s \\ ||S_p - T_s||_1 & \text{if } S_p \geq T_s \end{cases} \tag{5}$$

The properties we restrict are (1) the arc length of the control polygon $S_l = ||\mathbf{p}_1 - \mathbf{p}_2|| + ||\mathbf{p}_2 - \mathbf{p}_3||$, where $\mathbf{p}_i = [x_i, y_i]^T$ are the three control point locations, and (2) the center stroke radius $S_w = (r_1+r_2)/2$, where $r_1$ and $r_2$ are the start and end radii. Theses losses are summed over all canvases, i.e., $\mathcal{L}_{\text{style}} = \sum_t \mathcal{L}_{\text{shape}}(I, C_t)$ in this case.

Figure 4(c), Figure 6(left: b,c,d) and Figure 7(c,g) show artistic styles using these losses. The second row (b) in Figure 6(left) shows a style using an arc length threshold $T_s = 0.3$. Strokes longer than a third of the canvas are penalized, and so the network compensates with thicker strokes. This results in a smooth image approximation. As shown in the third row (c), penalizing thick strokes ($T_s = 0.05$) leads to a style with few large strokes in low-detail regions, instead placing much more detail in the foreground. Due to the difficulty in approximating the input image with 300 thin strokes, the network assumes a larger cost in a small number of strokes on low-frequency areas of the input image by choosing thicker strokes, resulting on an interesting contrasting style of thick and thin strokes. The fourth row (d) combines both these constraints. For these experiments, we use $D_{LSTM}$ and we set $\lambda_1$ to 0.01, $\lambda_2$ to 1 and $\lambda_3$ to 0.001.

**Noisy Motions.** With enough drawing skills and time, a human can accurately paint a precise painting of a target image. However, on a limited time budget, painting becomes less precise and more like a rough sketch. These rapid motions create their own styles in writing (Berio et al., 2016) and painting.

We approximate the effect of motion dynamics given by a limited time per individual stroke by adding Gaussian noise into the actions. Let $\mathbf{p}_{1:3}$ be a vector of brushstroke coordinates produced by the painter network at run-time. We add noise to the control points as $\hat{\mathbf{p}}_{1:3} = \mathbf{p}_{1:3} + z$ where $z \sim \mathcal{N}(0, \beta^2 \mathbf{I})$. The

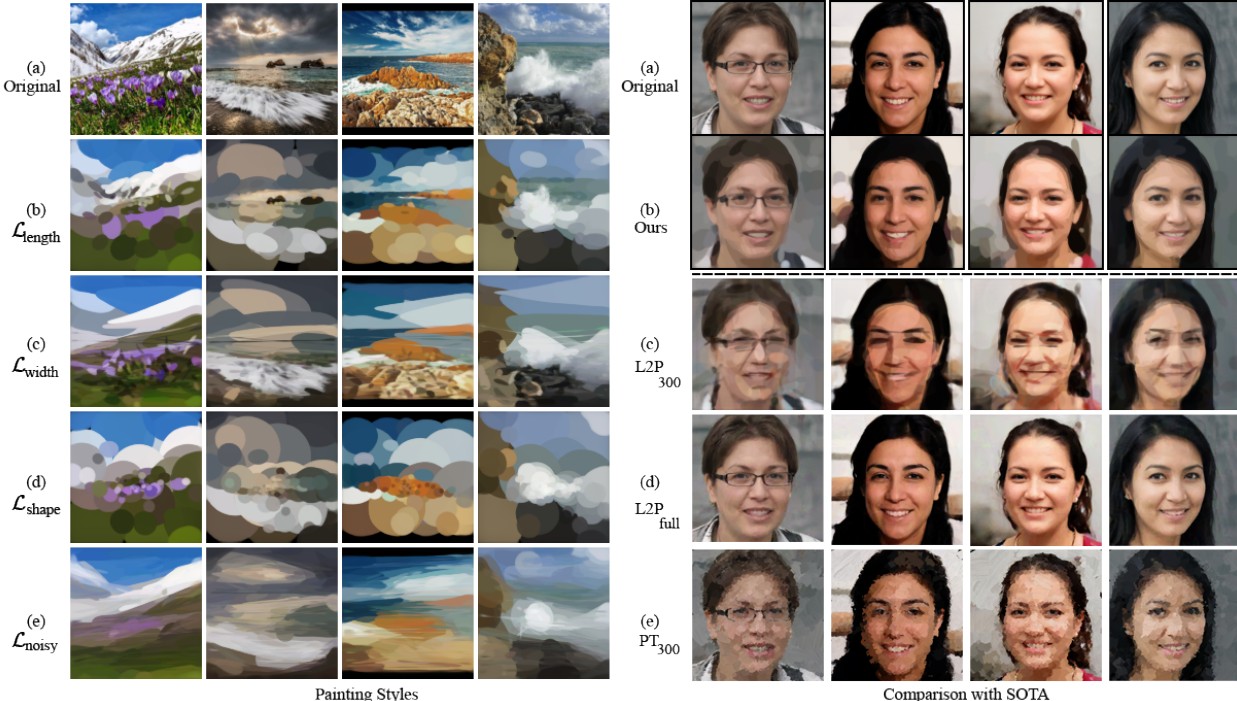

Figure 6: Painting Styles and SOTA Comparison (Left) (a): input image. (b): style by length constraint on stroke length. A small penalty encourages the network to approximate the image with short strokes. The network generally chooses a larger stroke thickness to balance the lack of long strokes. (c): style by stroke width penalty. The network learns to quickly approximate the upper part of the image with very few strokes that receive a penalty, creating a detail foreground and coarse background effect. (d): a combination of constraints on width and length leads to a style of thick and short rounded shapes in a coarse but smooth style. (e): scratchy effect by approximating rapid motions with Gaussian noise. The photos here are taken from the training set. We use $D_{LSTM}$ for all styles. (Right) (a): Original. (b): Our results using $D_{LSTM}$. (c) and (d): Learning-to-Paint Huang et al. (2019) limited to 300 strokes and full model, respectively. (e): PaintTransformers Liu et al. (2021) limited to 300 strokes.

minimization objective becomes

$$\mathcal{L}_{\text{noisy}} = \lambda \mathcal{L}_{\text{perc}}(I, C_T) + \sum_{t=1}^{T} \mathbb{E}_{z_t \sim p_z} \left[ \mathcal{L}_{\text{guidance}}(I, C_t) \right] \tag{6}$$

where $C_t$ is affected by the noise value as described above. In the context of stochastic optimization, we can approximate the expectation with a single random sample, and, further improve optimization by fixing the random seed (Spall, 2003; Ng & Jordan, 2000). We use $D_{LSTM}$ and set the same lambdas as for the precise network, and we use $\beta = 0.05$ and $\lambda = 0.01$. We do not apply noise at test time.

Figure 4(d), Figure 6(left: e) and Figure 7(d,h) show examples of this style. Because the network learns that movements are noisy, the painter overrides any detail and approximates the main image composition with scratchy strokes.

## 5 Comparison with State-of-the-Art Methods

We compare our method with three previous works: an optimization based method, Stylized Neural Painting (SNP) by Zou et al. (2021), an RL method, Learning to Paint (l2p) by Huang et al. (2019), and a stroke prediction method based on transformers by Liu et al. (2021). Simply comparing reconstruction error alone

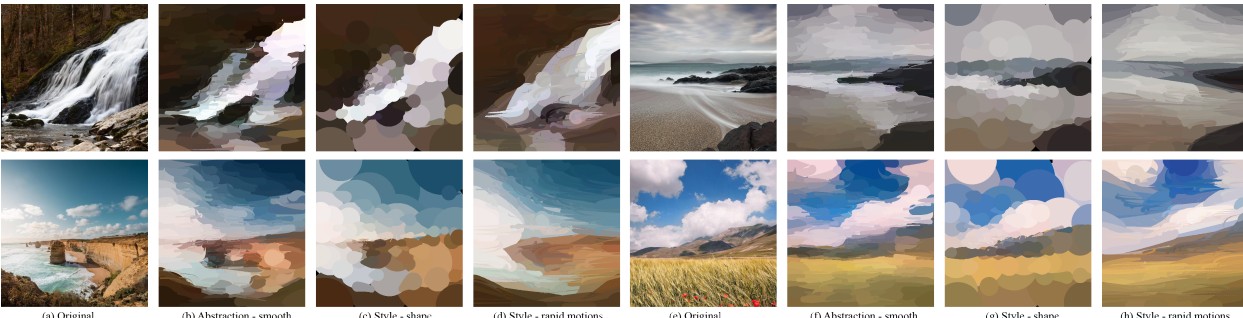

| (a) Original | (b) Abstraction - smooth | (c) Style - shape | (d) Style - rapid motions | (e) Original | (f) Abstraction - smooth | (g) Style - shape | (h) Style - rapid motions |

Figure 7: 512x512 300-stroke paintings. These examples illustrate results with higher resolution, and show how our method can provide nuanced, interpretable control over styles through varying losses and other interpretable factors. (a,e) original. (b,f) shows a basic style by smooth visual abstraction. (c,g) adds constraints on thickness and length that reduces the number of small strokes. (d,h) adds random noise to strokes resulting in a scratchy style.

| Method | Model-300S | | | Full Model | | |
|---|---|---|---|---|---|---|
| | $\mathcal{L}_1 \downarrow$ | $\mathcal{L}_{perc} \uparrow$ | N. Strokes | $\mathcal{L}_1 \downarrow$ | $\mathcal{L}_{perc} \uparrow$ | N. Strokes |
| l2p (RL) Huang et al. (2019) | 0.037 | 0.729 | 300 | **0.022** | **0.890** | 1000 |
| Paint Transformers Liu et al. (2021) | 0.089 | 0.490 | 351 | 0.057 | 0.609 | 3212 |
| SNP (Optimization) Zou et al. (2021) | 0.043 | 0.631 | 300 | 0.030 | 0.625 | 1000 |
| Ours$_{D_{LSTM}}$ | **0.035** | **0.747** | 300 | - | - | - |
| Ours$_{D_{FC}}$ | 0.046 | 0.735 | 300 | - | - | - |

Table 1: Quantitative results on 500 random faces from the CelebA dataset. We limit each method to 300 strokes the painting algorithms from previous work. We measure $L_1$ loss and perceptual loss using cosine similarity between input image and painting feature maps. The existing methods can achieve lower perceptual loss in their full models, but these models use thousands of tiny strokes and do not produce painterly abstraction.

would provide a poor metric for painting algorithms. In this case, the best scores would be achieved by an algorithm that produces thousands of tiny strokes, reproducing the input image almost exactly, but not looking much like a painting. Instead we focus on reconstruction error, not as a measure of image quality but a measure of economy. How well can an algorithm reconstruct an image with only 300 brush strokes?

**Quantitative Comparison** We first provide a comparison on portraiture with existing methods on our precision network, trained on CelebA, and reported in Table 1. In the first two columns all models use around 300 strokes. SNP is an optimization method and to ensure a fair comparison, we limit the number of strokes in the optimization procedure. Paint Transformer casts the painting process as a stroke prediction task, and generates a set of 8 random background strokes in a 32x32 canvas, and a second set of 8 foreground strokes. The network is tasked to correctly guess the foreground stroke parameters from the background set of strokes. Since it trains on a random number of strokes, the training process is not related to the final number of strokes used at inference. To ensure a fair comparison, we adjust the algorithm so that the total number of strokes is approximately 300. We retrain l2p with 300 strokes to ensure fair comparison. Our high precision network outperforms previous methods when using 300 strokes. Our $D_{LSTM}$ achieves better reconstructions than $D_{FC}$.

Table 2 shows quantitative results on landscapes using 300 strokes (left), and 1000 or more strokes (right). Metrics are calculated using 100 samples from a held-out set. We use $D_{LSTM}$ decoder. Our method achieves better scores in perceptual loss than previous methods, but does not obtain better metrics than l2p on pixel loss. We add our method with fine-tuning step for information purposes.

| Method | Model-300S | | | Full Model | | |
|---|---|---|---|---|---|---|
| | $\mathcal{L}_1 \downarrow$ | $\mathcal{L}_{perc} \uparrow$ | N. Strokes | $\mathcal{L}_1 \downarrow$ | $\mathcal{L}_{perc} \uparrow$ | N. Strokes |
| l2p (RL) Huang et al. (2019) | **0.041** | 0.651 | 300 | **0.040** | **0.814** | 1000 |
| Paint Transformers Liu et al. (2021) | 0.092 | 0.513 | 351 | 0.068 | 0.678 | 3212 |
| SNP (Optimization) Zou et al. (2021) | 0.055 | 0.561 | 300 | 0.046 | 0.561 | 1000 |
| Ours$_{D_{LSTM}}$ | 0.048 | **0.658** | 300 | 0.045 | 0.667 | 1000 |
| Ours$_{D_{LSTM}+FT}$* | 0.023 | 0.868 | 300 | 0.020 | 0.899 | 1000 |

Table 2: Quantitative results on 100 random sample images drawn from a held-out set on landscapes. We provide metrics on a 300-stroke model (left), and a 1000-stroke model (right). For Liu et al. (2021), total number of strokes is averaged over the held-out set. We measure $L_1$ loss and perceptual loss using cosine similarity between input image and painting feature maps (higher is better). For a fixed number of strokes, our method achieves a lower perceptual loss than previous methods. *We include the optional finetuning step for information purposes; a more meaningful comparison is done without such finetuning step.

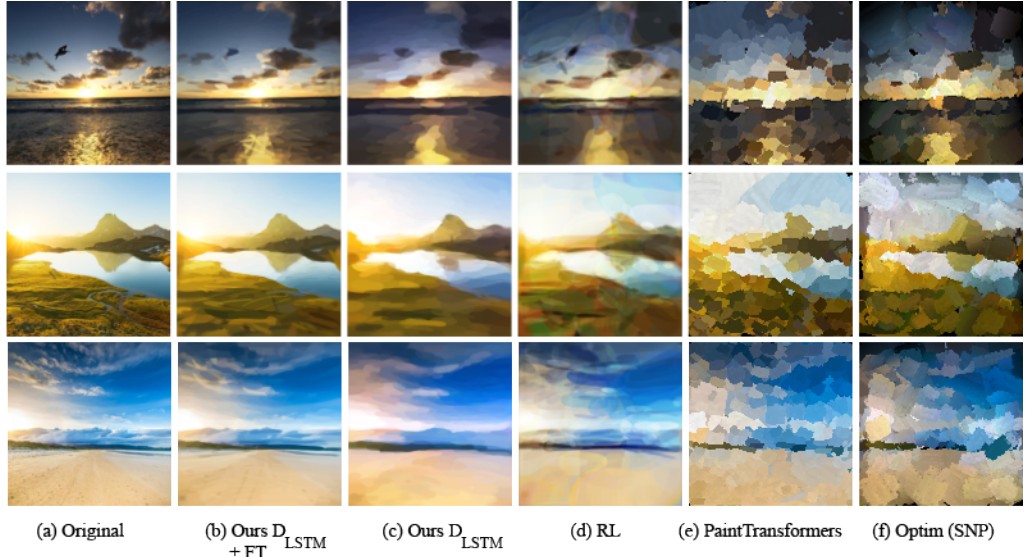

(a) Original    (b) Ours D$_{LSTM}$ + FT    (c) Ours D$_{LSTM}$    (d) RL    (e) PaintTransformers    (f) Optim (SNP)

Figure 8: Precise reconstructions comparison with previous methods. All methods use 300 strokes. Our method with a fine-tuning step (a) is able to reconstruct the input image better. Without fine-tuning step (b) our method still outputs smoother paintings than previous work. Paint Transformers (Liu et al., 2021) and SNP (Zou et al., 2021) output very large strokes and they suffer with reduced stroke budgets.

**Qualitative Comparison.** The lack of previous methods that produce visual abstractions and styles without the use of external style images or brush textures, challenges an assessment on styles. We instead compare our precision network with existing methods. Our network is able to produce crisper and more defined faces in far fewer strokes. Learning-to-paint (Huang et al., 2019) achieves better reconstructions with hundreds of more strokes as seen in Figure 6(right). This figure compares our method with RL (Huang et al., 2019) and Transformers (Liu et al., 2021). Our results are shown in the second row (b). (c) and (d) show the RL approach with 300 and full model, respectively. (e) shows Paint Transformer with 300 strokes. We see that our model places strokes more efficiently than previous methods.

Figure 8 shows a qualitative comparison of our networks with previous work. Our network without fine-tuning step seems to maintain a more consistent use of colors, and the paintings are slightly less noisy than the RL method by Huang *et al.* Huang et al. (2019) (Figure 8 (d)). Paint Transformer Liu et al. (2021) (Figure 8 (e)) and Stylized Neural Painters Zou et al. (2021) (8 (f)) share the same stroke model, resulting

in a coarser representations and needing many more strokes to produce accurate representations of the input image.

## 6  Ablation Study

**Mapping Function.**  We run ablations on two types of mapping functions, a direct mapping or projection from CNN feature maps to a sequence of strokes, $D_{FC}$, and a sequential mapping that consists on a first linear layer to map CNN feature maps to a sequence of hidden vectors, followed by LSTM and FC layers, $D_{LSTM}$. We show results of both models in the appendix, and provide insight for the size of such mapping function. We found that for visual abstractions, the LSTM-based architecture does not improve our results. However, we obtain our best results for precise reconstructions, reported on Table 1 on CelebA, and styles using our $D_{LSTM}$ model. $D_{FC}$, however, is sensible to the number of parameters.

**Loss Functions and Optimization Regime.**  There is a relationship between the effect of a particular loss and the optimization regime. When opting for a greedy optimization, doing perceptual loss at each intermediate canvas becomes computationally expensive. However, by adding perceptual loss at the last time step the paintings achieve a better approximation to the input image. We found that when adopting less strict optimization regimes, such as evaluating the loss at the last frame, the impact of the perceptual loss decreases drastically. This is because the lack of backpropagated guidance disables the network to paint with more detail. We present an extensive ablation on loss functions in appendix, where we describe other losses and training approaches such as adversarial training with Wassertein loss, or contrastive loss.

## 7  Conclusions

This paper presents an approach for economical-style paintings. We argue that *economy* is a foundational approach for many painting styles and a step towards developing general-purpose painting techniques. We see that our method, in general, places strokes more efficiently than previous methods, which, in contrast, spray thousands of strokes to reproduce the input image, which leads to a less desirable approach to work on styles.

**Limitations.**  Our algorithm still suffers from local minima, and the optimization for the precision network converges slowly, resulting in larger training time. We choose to work with a tight stroke budget of 300 strokes to produce economical paintings, which we demonstrate enough for good reconstructions of size 128x128. Since the algorithm outputs stroke parameters, at test time we can substitute the pretrained neural renderer by a non-differentiable renderer and make the painting larger, as seen in Figures 4 and 7. Even though our best results on precision on CelebA do not need further processing, in order to achieve accurate reconstructions on landscape images we need to do a fine-tuning operation. This is partially because the variance in the distribution of landscapes images is much higher than the CelebA dataset. This step is not needed for any other style.

**Discussion and Future Work.**  Our work scratches the surface for economical painterly stylization. Previous work focuses either on good reconstructions of an input, or on generating some level of abstraction via texture maps or relying on external style images. For instance, l2p (Huang et al., 2019) is able to approximate the input with high accuracy with more than a thousand strokes, but does not output any stylistic variation. Paint Transformer (Liu et al., 2021) and SNP (Zou et al., 2021) output paintings in a very similar style. They use texture maps in their strokes and SNP achieves style by changing the stroke primitive or doing style transfer. Our work, however, focuses on the generation of variations and economical styles while keeping the same stroke primitive (quadratic Bézier curve). The concept of economy in art opens up new stylistic possibilities, and we design our networks under this scenario. We find it intriguing that we can obtain competitive results with a direct mapping approach, finding an alternative to more complex approaches such as RL, Transformers, or semantic guidance models. Future work comprises a better understanding of a direct mapping from convolutional features, finding a larger spectrum of style variations, and exploring a more controllable path to output strokes.

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

# A Appendix

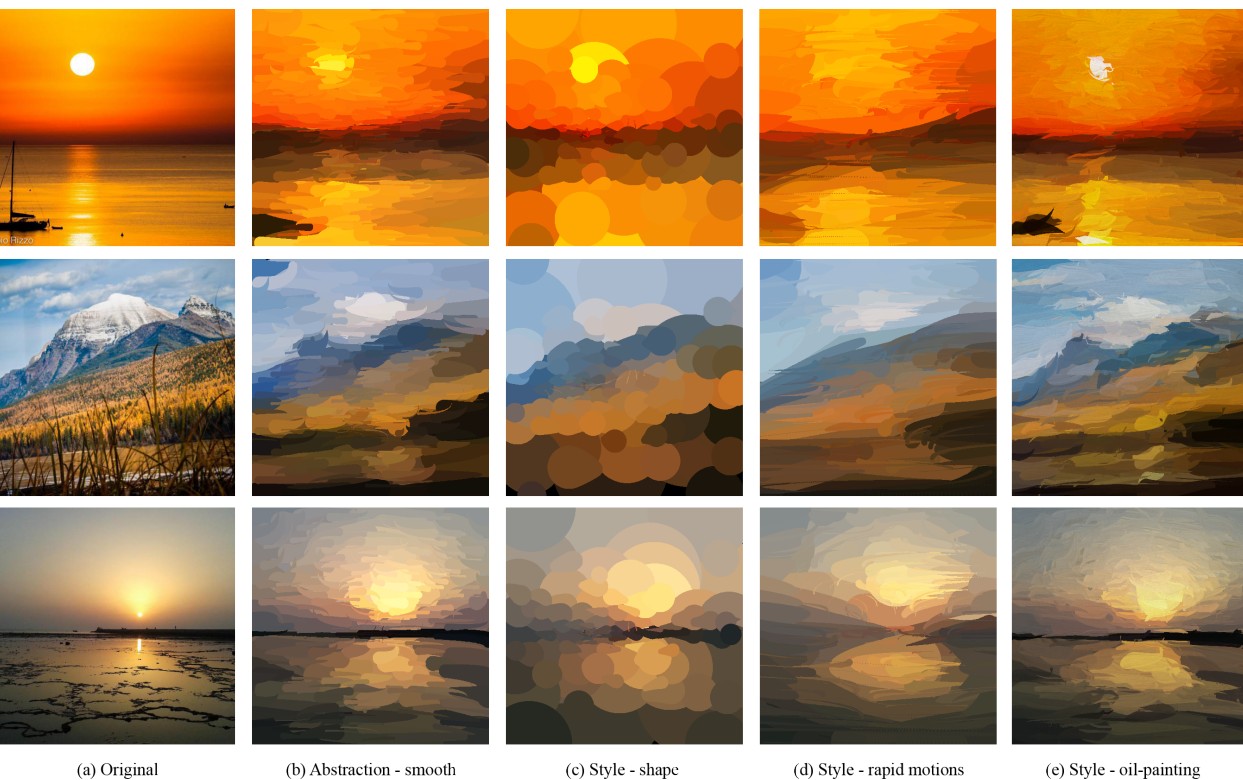

|  |  |  |  |  |
|---|---|---|---|---|
| (a) Original | (b) Abstraction - smooth | (c) Style - shape | (d) Style - rapid motions | (e) Style - oil-painting |

Figure 9: 512x512 300-stroke economical paintings. These examples illustrate results with higher resolution, and show how our method can provide nuanced, interpretable control over styles through varying losses and other interpretable factors. (b) shows a basic style by smooth visual abstraction. (c) adds constraints on thickness and length that reduces the number of small strokes. (d) adds random noise to strokes resulting in a scratchy style. (e) oil-painting style.

## A.1 Higher Resolution Paintings

We show 512x512 visual abstraction and styles landscapes paintings with 300 strokes (Figure 9 and **??**) by using the set of strokes parameters output by the network and rendering with a non-differentiable renderer

| Method | $\mathcal{L}_1 \downarrow$ | $\mathcal{L}_{perc} \uparrow$ | Num Strokes |
|---|---|---|---|
| AbstNet $D_{FC-S}$ | 0.038 | 0.738 | 300 |
| AbstNet $D_{FC-L}$ | 0.043 | **0.750** | 300 |
| Precise-Style-Net $D_{LSTM}$ | **0.035** | 0.747 | 300 |

Table 3: Quantitative results on same held-out set on CelebA as in Table 1 in main paper, using networks explained in Figure 10. Precise-Style network achieves a much lower $L_1$ score than AbstNet. The latter, however, achieves a slightly better perceptual score. However, as shown in Figure 14(b-c), it produces slightly worse reconstructions.

at a higher resolution. Figure 9 and **??** (b) show smooth abstraction using pixel loss at all intermediate canvases. Figure 9 and **??** (c) show artistic style with constraints on thickness and length. Figure 9 and **??** (d) show artistic style given by rapid motions. Figure 9 and **??** show that our method provides a very fine-grained and interpretable control with the potential to expand the styles more, as opposed to previous work that have no control over nuances and details of their paintings. This is further shown in Figure 8.

## A.2   Ablations on Model Architecture

This section shows ablations on both presented networks: visual abstraction net, based on FC layers in decoder, and precise and style net based on FC and LSTM layers in decoder. For simplicity, we name the former AbstNet (which uses $D_{FC}$) and the latter Precise-Style-Net (which uses $D_{LSTM}$). Note that Precise-Style-Net is used for precise reconstructions (Figure 1(b) and Figure 5(left)) as well as artistic styles (Figure 1(f-h), Figure 4(c-d) and Figure 6(left)), while AbstNet is only used for visual abstractions (Figure 1(c-d), Figure 4(b), Figure 5(right), Figure 7(b,f)). We first ablate on network sizes and encoder-decoder combinations for our AbstNet.

The left part of Figure 10 (a-c) shows the full architecture of three abstraction networks, ordered with increasing number of parameters, from left to right. Figure 10 (d) shows the full architecture of our Precise-Style-Net, which has ~90M learnable parameters. We find a relation between number of learnable parameters and abstraction quality. Smaller networks with ~30M and ~50M parameters (Figure 10 (a),(b), respectively) are able to produce higher quality visual abstractions. We find that increasing the spatial size of the encoded feature map (a) doesn't affect the output painting significantly. However, reducing the number of parameters of the network increases the abstraction capability, while increasing the number of parameters Figure 10 (c) (~100M) results in better approximations to the input image and thus, reducing the level of abstraction. The result of these ablations are reflected in the following sections.

## A.3   Visual Abstraction

Figure 11 shows model ablations on the coarsest visual abstraction using objective $\mathcal{L} = ||I - C_T||_1$, where $C_T$ is the canvas at the last time step. All paintings are made with 300 strokes. Except for the bottom row, which uses Precise-Style-Net (Figure 10 (d)), the other rows use different versions of AbstNet, specifically, second row uses Figure 10 (a), third row uses Figure 10 (b), and fourth row uses 10 (c). Smaller networks produce better (coarser) abstractions than bigger networks, which is desirable for this objective function. We find that the architecture shown in Figure 10 (b) produces better results for the range of visual abstractions that we present in this work.

## A.4   Visual Abstraction Painting Progress

Figure 12 shows canvases rendered every 25 strokes. Top row shows visual abstraction given by applying $L1$ loss at the final frame. This corresponds to a one-off optimization schema, which encourages a non-greedy behaviour. As a result, the network wastes the majority of the strokes, achieving a coarse abstraction with the last 50 strokes. Abstraction in middle row is achieved using the loss defined in Equation (4) (medium). Specifically, $\sum_{t=1}^{T/k} ||I - C_{kt}||_1$, where $T/k$ is an integer, $k = 6$, and it corresponds to a sparse optimization schema, backpropagating errors every 50 strokes. Here only the first few strokes are wasted. The bottom

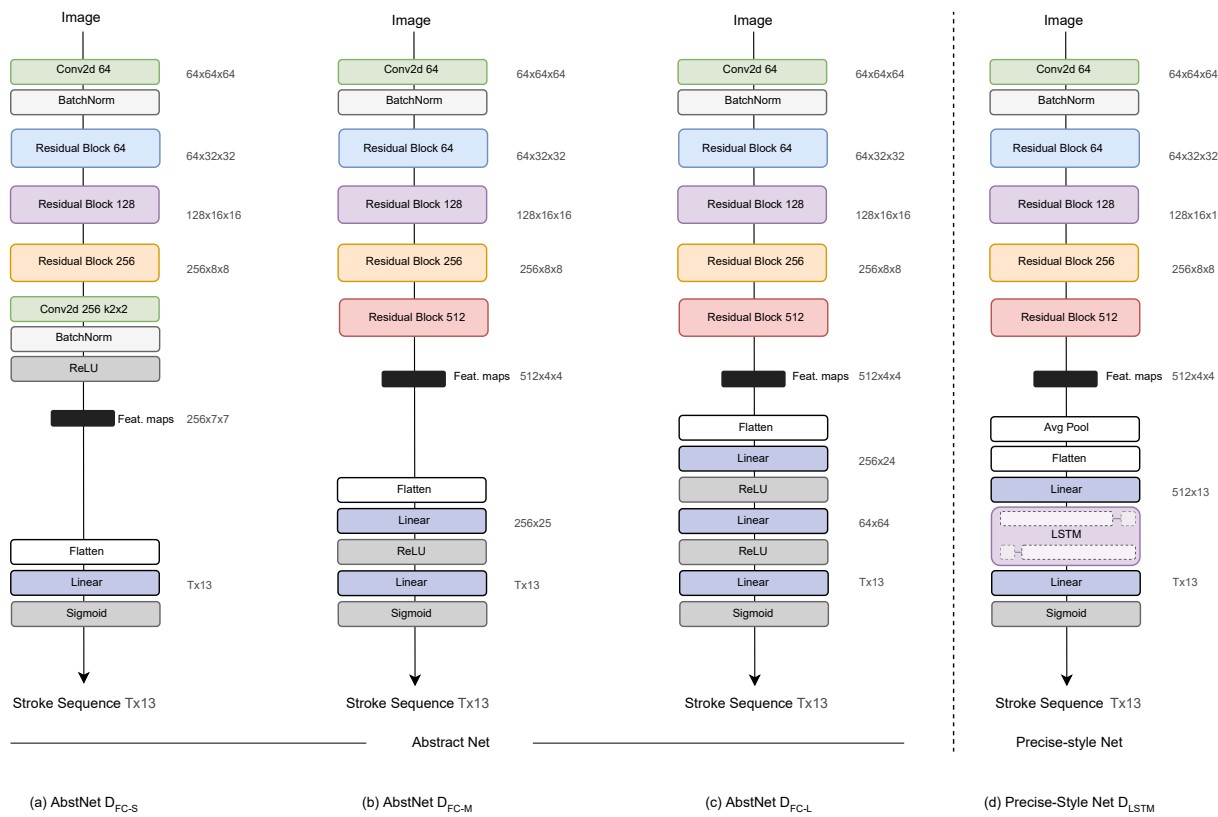

Figure 10: Model Ablations. Ablation on network architectures for visual abstractions for 300 strokes (a-c) and precise-style network (d). (a) Encoder uses the first 3 residual blocks of Resnet-18 and an additional convolutional layer to extract a larger feature map. Decoder is composed by 1 linear layer followed by non-linear activation. Learnable parameters: ∼31M. **(b) Abstract net used in this paper**. Encoder uses the first 4 residual blocks of Resnet-18. Decoder has 2 linear layers followed by non-linear activations. Learnable parameters: ∼50M. (c) Encoder is the same as (b). Decoder is composed by 3 linear layers followed by non-linear activations. Learnable parameters: ∼100M. (d) Precise-Style network, used for precise reconstructions and styles. Decoder is composed by a linear layer, followed by a bidirectional LSTM, followed by a final linear layer. Learnable parameters: ∼90M.

row corresponds to a more guided regime which backpropagates the error at every time step. This results in a greedy behaviour and no strokes are wasted.

## A.5   Artistic Styles

Figure 13 shows model ablations on artistic style given by constraining both stroke thickness and length. All paintings are made with 300 strokes. Figure 13 (b) shows the results of this style using our Precise-Style network. This network produces clearer results, defining object boundaries better than columns (c) and (d). Even though column (b) and (c) use networks with a similar number of parameters, the LSTM layer seems provide a better structured sequence of strokes. This behaviour happens across all artistic styles and for precise reconstructions.

## A.6   Precise Representations on Portraits

Figure 14 demonstrates the improvement that Precise-Style net has over AbstNet on precise reconstructions on CelebA. We compare our Precise-Style LSTM-based network with two abstract FC-based networks, one

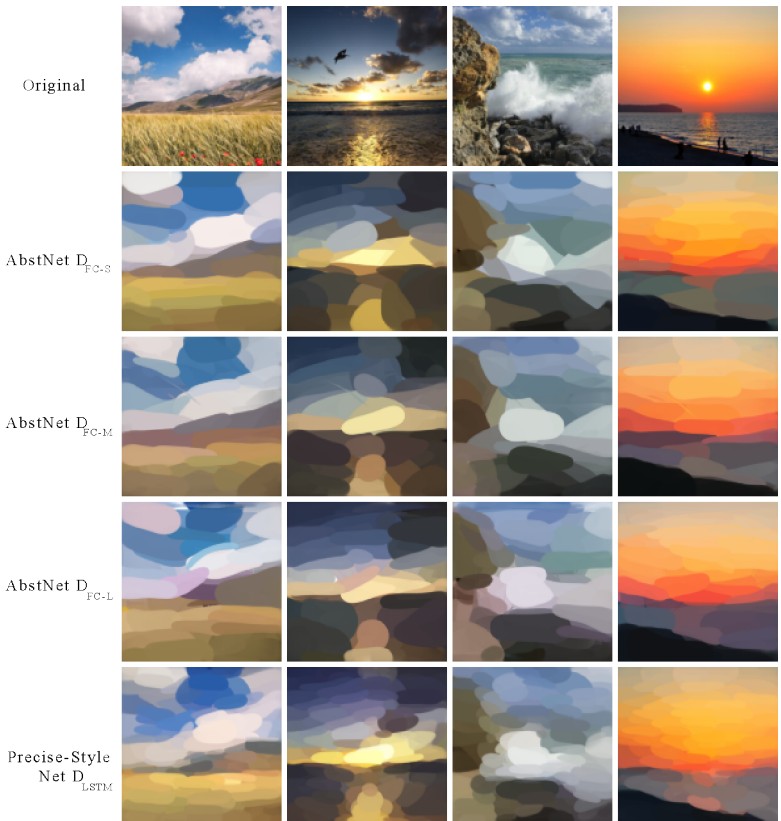

Figure 11: Model ablations on visual abstraction. All visual abstraction paintings are made with 300 strokes. Top row: original image. Second row: AbstNet using architecture shown in Figure 10 (a). Third row: AbstNet using architecture shown in Figure 10 (b). Fourth row: AbstNet using architecture shown in Figure 10 (c). Bottom row: Precise-Style-Net (Figure 10 (d)). The goal of this method is to provide a coarse level of abstraction, and we find a relation between network size and level of abstraction. We see how second and third rows are better abstractions than fourth and last rows. There is not a substantial difference between second and third rows, and both are suitable for this abstracted style. For all visual abstractions shown in the paper, we choose to work with the network corresponding to the third row (10 (c)).

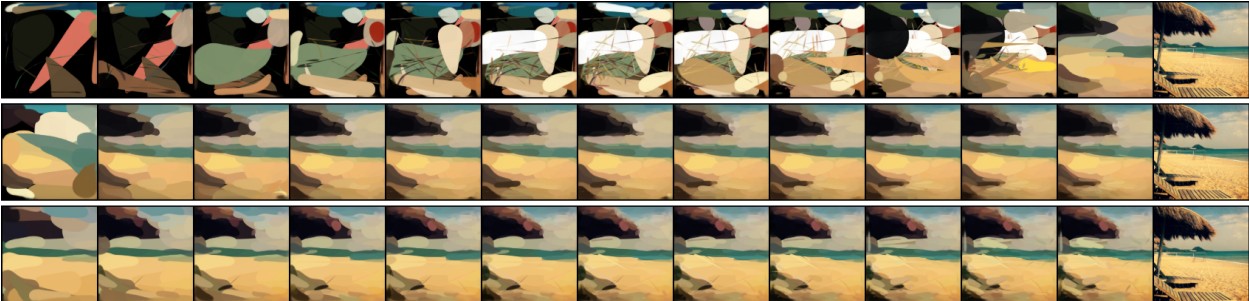

Figure 12: Visual abstraction painting progress of different optimization schemas. All three rows use 300 strokes; intermediate canvases are displayed every 25 strokes. Top row: applying the loss only on the last canvas. The network "wastes" the majority of the strokes, with few of them contributing much to the final painting. Middle row: applying the loss only on every 50th canvas. Bottom row: applying the loss at every canvas, as in Equation 3 in main paper.

with approximately the same number of parameters than our Precise-Style net (∼100M parameters, Figure 10 (c)), and a smaller network with ∼30M parameters (Figure 10 (a)). We see how the precise net outputs sharper paintings (Figure 14 (d)) because it is able to capture finer details, such as main facial wrinkles, and expressions. This is perhaps more noticeable in rows 3 and 4. The paintings given by the smallest network, shown in Figure 14 (b), are able to maintain the overall structure and composition, however, produce coarser paintings. Figure 14 (c) improves upon (b) but fails to accurately eyes and and facial gestures. Metrics corresponding to this study are reported in Table 3. Even though the larger AbstNet gives a slightly better perceptual score, overall the image is better reconstructed with Precise-Style net.

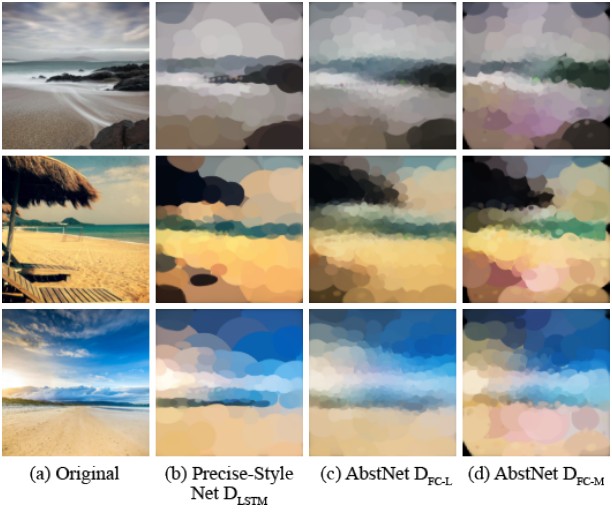

(a) Original    (b) Precise-Style Net $D_{LSTM}$    (c) AbstNet $D_{FC-L}$    (d) AbstNet $D_{FC-M}$

Figure 13: Model ablations on styles. All paintings are made with 300 strokes. (a) Original. (b) Style produced by our Precise-Style-Net. (c) AbstNet with a similar number of trainable parameters than (b). (d) AbstNet with approximately half of the total number of trainable parameters than the previous two. (b) shows a clear style, defining object boundaries much better than our abstract net, which outputs blurrier paintings.

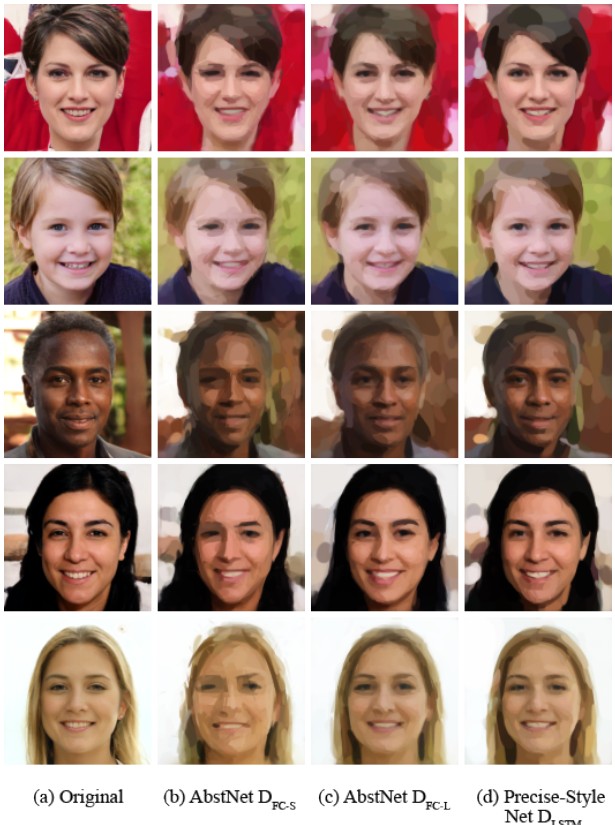

(a) Original    (b) AbstNet $D_{FC-S}$    (c) AbstNet $D_{FC-L}$    (d) Precise-Style Net $D_{LSTM}$

Figure 14: Model ablations on reconstructions. This figure compares results on precision on StyleGAN generated faces obtained by using three different architectures using 300 strokes. Column (b) uses AbstNet with one linear layer (Figure 10 (a)). Column (c) uses AbstNet with 2 linear layers (Figure 10 (c)). Column (d) uses our Precise-Style-Net with decoder $D_{LSTM}$. All of them capture overall composition and facial gestures. However, (b) produces coarse strokes leaving some facial features such as eyes undefined, and color gradients are not smooth. (c) improves upon (b) but is unable to capture finer details such as facial gestures or eyes. (d) results in a sharper reconstruction, capturing finer details and edges better.

## A.7 Stroke Budget Ablations

This section shows results a comparison of our precise network with 300, 600 and 1000 strokes in Figure 15 columns (b), (c) and (d), respectively. Increasing the stroke budget while maintaining the resolution results in blurrier paintings. We report the metrics for the 300-stroke and 1000-stroke models in Table 2. Metrics show that the 1000-stroke outperforms the 300-stroke model, however, the quality of the 1000-stroke painting is much worse than the 300-stroke painting, which indicates that other metrics might be more suitable for painting algorithms.

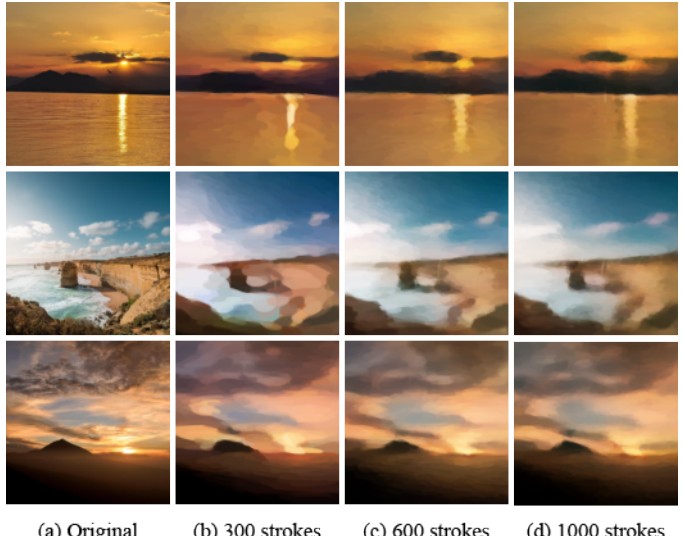

(a) Original  (b) 300 strokes  (c) 600 strokes  (d) 1000 strokes

Figure 15: Stroke budget ablations. Precise-Style-Net using (b) 300 strokes, (c) 600 strokes and (d) 1000 strokes. Adding more strokes produces a blur effect on the painting.

## A.8 Loss Functions

Different loss functions in combination with different decoders, $D_{FC}$ and $D_{LSTM}$ give different levels of control of the painting, as shown in previous sections. In this section we show two the effect of two other loss functions.

### A.8.1 WGAN-GP Loss

Following Mellor et al. (2019); Ganin et al. (2018); Huang et al. (2019); Liu et al. (2021), we also explore the impact of an adversarial loss for precise paintings. Specifically, we add the same discriminator network, $D_{net}$, as Huang et al. (2019), and use WGAN-GP (Gulrajani et al., 2017) in our loss function. $D_{net}$ discriminates between real pairs (input image, input image) $Q$, and fake pairs (input image, painting) $P$. WGAN-GP loss is:

$$\mathcal{L}_{WGAN-GP} = \mathbb{E}_{P\sim\mathbb{P}}[D_{net}(P)] - \mathbb{E}_{Q\sim\mathcal{Q}}[D_{net}(Q)] \\ + \lambda\mathbb{E}_{\hat{Y}\sim\hat{y}}[(||\nabla_{\hat{Y}}D_{net}(\hat{Y})||_2 - 1)^2] \tag{7}$$

where $P$ is the pair canvas-input image at the last time step, $Q$ is the pair input image-in put image, and $\hat{Y}$ is sampled from $Q$ and $P$ with $t$ uniformity sampled between 0 and 1, $\hat{Y} = tQ + (1-t)P$ with $0 \leq t \leq 1$. We substitute perceptual by WGAN-GP loss in our general formulation, and thus, our loss function becomes:

$$\mathcal{L}_{\text{precise}_{\text{WGAN-GP}}} = \lambda_1 \sum_{t=1}^{T} \mathcal{L}_{pixel_t} - \lambda_2 \mathcal{L}_{WGAN-GP} \tag{8}$$

We set $\lambda_2 = 0.01$. We find that adopting an adversarial schema does not improve perceptual losses, as shown in Figure 16 (c). In some cases, it introduces some artifacts that make the painting a bit noisier, as seen in Figure 16 bottom row. We hypothesize that this is partially due to the gap between distributions of real images and distributions of 300-stroke paintings.

### A.8.2 Contrastive Loss

A contrastive approach might seem suitable for precise paintings. Same as in the previous section, we remove perceptual loss and keep pixel loss. The goal of this loss is to minimize the cosine distance of a vector representation of the input image $\vec{v}$ with a vector representation of the canvas $C_T$ $\vec{w}$, while maximizing the distance between $C_T$ $\vec{w}$ and a set of negative samples $Z = \{\vec{z_1}, \vec{z_2}, ..., \vec{z_k}\}$. Contrastive loss is calculated as :

$$\mathcal{L}_{contrastive} = -log\frac{exp(dist(\vec{w}, \vec{v})/\tau)}{\sum_{k=1}^{K} exp(dist(\vec{w}, \vec{z_k})/\tau)} \tag{9}$$

We set $K = 10$ and randomly sample $K$ images from the training set where $k \neq i$, and $i$ is the index of the vector representation $\vec{v}$ corresponding to the positive sample. Our loss function becomes:

$$\mathcal{L}_{\text{precise}_{\text{contrastive}}} = \lambda_1 \sum_{t=1}^{T} \mathcal{L}_{pixel_t} - \lambda_2 \mathcal{L}_{contrastive} \tag{10}$$

We set $\lambda_2 = 0.0001$ for our ablation. Figure 16 (d) shows the paintings obtained using Equation (10). As we see, this loss does not improve results using perceptual loss, and results are inconclusive. We hypothesize that a more robust contrastive learning approach with more data augmentations or robust architectures as done in He et al. (2020).

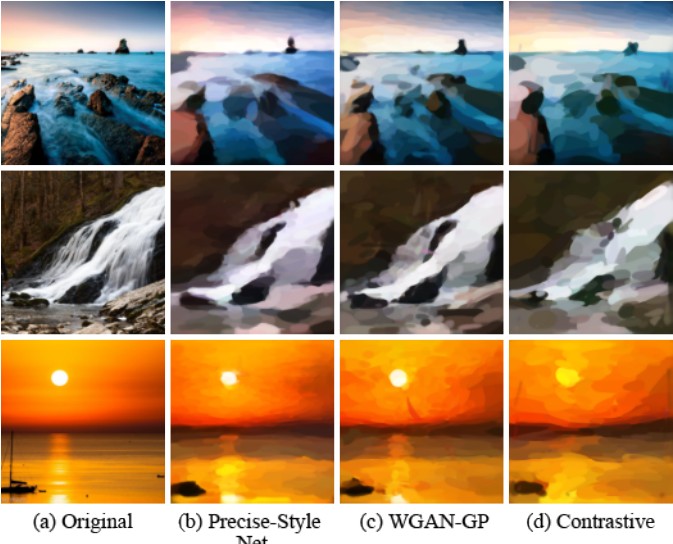

(a) Original     (b) Precise-Style Net     (c) WGAN-GP     (d) Contrastive

Figure 16: **Effect of using WGAN-GP and contrastive losses**. We substitute perceptual loss by WGAN-GP (c) and contrastive loss (d) following Equation (9).

## A.9 Stroke Optimization

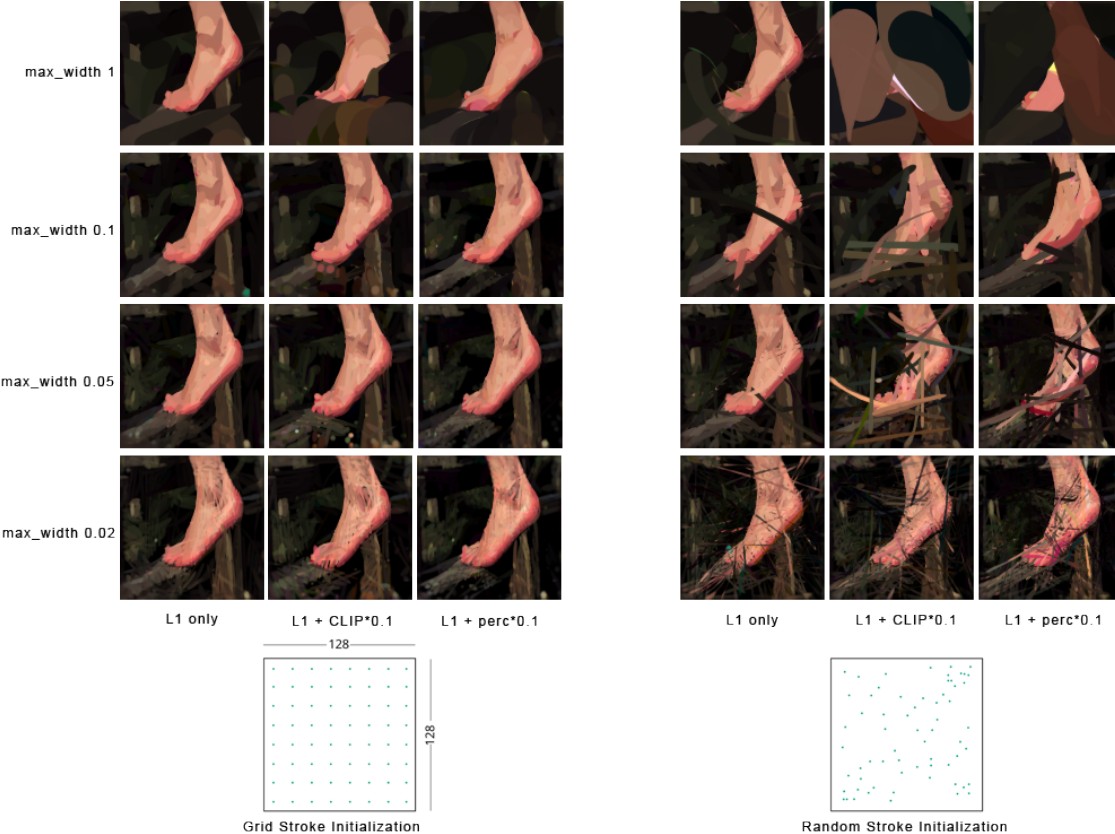

Figure 17: Stroke initialization ablation.

We show some experiments on direct stroke optimization. Initializing a random set of strokes $S$ does not lead to good results. Figure 17 shows the difference between a random initialization (right) and a grid-based initialization based on heuristics (left). For the latter, we first lay out uniformly a $\sqrt{T}$ number of $x_1, y_1$ coordinates corresponding to the mid point $ct_1$ of the stroke across each of the $x_p, y_p$ dimensions of the canvas; with $T = 324$. For start and end points, we sample from a Gaussian distribution with mean $ct_1$ and standard deviation 0.1, which limits the length of each stroke at the initialization step. The rest of the parameters of each stroke are drawn from a uniform normal distribution. Here, we also test the impact of different losses. Middle column on both sides of the figure uses the pretrained CLIP model as equivalent to a perceptual loss. Right-most column uses the same perceptual loss as in the main paper.

We see how initializing random strokes does not work well, and further work needs to be done to achieve stylization (instead of reconstructions) using direct optimization.

## A.10 Oil Painting Comparison

We use the thin plate spline (TPS) algorithm to compute the mapping of a bitmap of an oil painting texture to the shape of our strokes. To obtain the control points that are fed into the TPS algorithm, we first divide the original Bézier curve into N parts and offset those points by their corresponding width to both sides. We then map an oil painting bitmap texture to the original shape of the stroke. Figure 18 shows a comparison between our approach (top) and the oil painting filter on Adobe Photoshop (bottom). We see that the latter smooths the curves slightly more than the TPS algorithm. The former maintains the original thinner strokes with more fidelity than the oil painting filter on Photoshop.

(a)

(b)

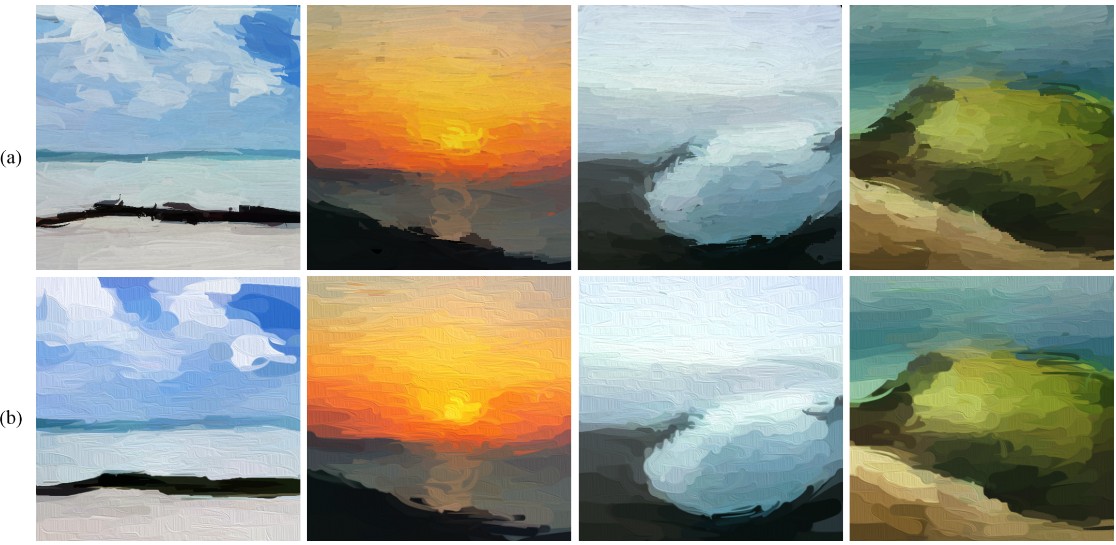

Figure 18: Comparison of oil painting texture via thin plate spline algorithm (top) and oil painting filter in Adobe Photoshop (bottom)

