# OpenReview forum: "Im2Painting: Economical Painterly Stylization "
_TMLR — Rejected by TMLR_

### Review · Reviewer_FXkV · 2022-06-28

**Summary Of Contributions:**

Authors propose a system that takes an image as input, and as output produces a set of 'paint strokes' that reproduce the image, either as precisely as possible, or in various more abstract (coarser) styles.

The work makes use of an existing parameterisation and differentiable renderer for paint strokes, based on Bézier curves. It operates by passing an image through an encoder CNN, then a decoder (recurrent or fully-connected) that yields a sequence of strokes. These components are trained jointly to reproduce input images, under a perceptual metric. The style of the images can be modified by constraining attributes of the strokes, e.g. limiting their length, width, or number.

The method is demonstrated on datasets of images showing human faces, and landscapes. When aiming for precise reconstructions, the method performs well on faces, and reasonably on landscapes (results are somewhat indistinct here). When aiming for 'artistic' reconstructions, results are reasonable on both datasets, though not particularly impressive. Quantitative experiments show the proposed method outperforms several baselines in terms of reconstruction quality under a fixed stroke budget.

**Broader Impact Concerns:**

None.

**Requested Changes:**

It would be interesting to see results on domain transfer, to see how specialised the learnt models are, including baselines (see above).

Establish and discuss what limits the fidelity of precise reconstructions (see above).

It would be interesting to see how well the proposed method works if the encoder/decoder neural networks are removed. In particular, the 'precise' landscape experiments already do a direct optimisation after applying the NN; one could instead initialise this randomly (or according to some sensible heuristic) and run for more iterations, and thus avoid the NN entirely.

It would be interesting to see a comparison with the Photoshop 'oil paint' filter, which is probably the actual practical baseline used by most designers interested in this task.

Each of the above would improve the paper, though I wouldn't describe any as 'critical' for acceptance. Also, all the following textual issues should be fixed:

The 'oil painting' style doesn't seem to be described anywhere – it just appears in a couple of result figures. Please add a description somewhere appropriate.

fig. 6 caption says it shows training-set images -- please clarify if this is the case for any other figures.

Start of Sec. 4 would benefit from some reorganisation -- it's titled " Economical Painterly Styles" and starts "In this section, we show a range of artistic styles", yet it fact the entire empirical results are in this section. It should be titled accordingly, and the introductory paragraph fixed to summarise all the results it contains. It might be sensible that the current "4.0" becomes 4.1, i.e. a subsection on different styles alongside that on comparison to SOTA. Alternatively, the styles could be discussed prior to the entire results section, but in that case without reference to the experimental results yet.

Similarly, sec. 4 'Training details' para – this should be earlier, probably at the start of sec 4.

At several points the loss is described as an energy function, but there isn't really any probilistic interpretation here (i.e. the energy defining a Gibbs distribution), so I think it'd be clearer just to refer to 'loss function' throughout.

p3 para 5: "...these methods generate thousands of strokes per image, with either more impressionistic effects or thousands of strokes" -- please fix this sentence (the repetition of 'thousands of strokes' doesn't make sense)

fig. 3 caption discusses "projection layer", but it's not clear which part(s) (the decoder?) this refers to in the diagram itself

p4 para 2: citations should be parenthesised (citep)

sec 3.2 para 1: "g consists [...] takes in the entire sequence of actions" -- I assume this means you apply g with each of the actions in turn, but the current wording is a bit misleading -- it sounds like the input to g's first layer is the entire sequence, whereas per my understanding, each stroke is passed in separately.

**Strengths And Weaknesses:**

Strengths:

The proposed method is fairly simple (compared with prior art using reinforcement learning, transformer architectures, etc.), requiring only a neural network encoder/decoder pair, and a differentiable renderer of brush strokes.

The pipeline is novel (to the best of my knowledge), although it is built from existing components (an existing differentiable stroke renderer, and standard neural networks).

Nonetheless, the method can successfully reproduce a variety of input images (faces and landscapes) up to 512x512 resolution, and its 'artistic' styles are able to re-render them in various interesting ways.

The empirical evaluation is fairly comprehensive, covering different 'styles', two datasets, and several baselines.  There are also various ablation experiments, showing the benefits of different aspects of the model, and justifying design decisions. Limitations are discussed.

The paper is well-written, and mostly well-structured (except sec. 4 – see below). Ideas are clearly motivated and described, and the text is readable throughout; there are very few typos and grammatical errors.

Weaknesses:

The paper is interesting, but does not make a great contribution to knowledge. It uses fairly standard components, assembled straightforwardly, and achieves reasonable results – but there is nothing especially innovative, nor any obviously transferrable insights for the reader.

The 'precise' reconstructions are not particularly precise – they are significantly coarser than the originals, most prominently for landscapes. It would be good to see an analysis of why this is the case – e.g. is it due to the expressivity of 300 strokes, or optimisation issues?

The 'artistic' reconstructions are not particularly artistic, in my (subjective) opinion. Compared with the impressionist styles used in the introduction to motivate the work, the results are somewhat disappointing. The 'oil painting' style (which appears in figures but lacks description anywhere) is also not very similar to actual oil paintings.

There are no experiments on more general images (e.g. ImageNet), nor on transfer (i.e. training the model on some restricted dataset – like those used currently – then applying it on a different domain). I think this is important to evaluate, as using a fully-learnt mapping from image to strokes is inherently less generalisable than doing a per-image direct optimisation, and it would be good to quantify this.

---

> ### Author Response · Authors · 2022-07-12
> **Response to Reviewer FXkV**
>
> Common response to all reviewers:
>
> We want to emphasize that _economy_ is not just “one style.” Many common painting styles involve considerable precision in their brush strokes, e.g., consider the way the examples in Figure 1 use a few strokes that are carefully aligned to image features, curves follow contours, and so on. Existing painting algorithms are not economical; they often spray the canvas with strokes, using many strokes to reproduce the appearance of the image well, but often with an “impressionistic” look as a result; one could not imagine capturing the styles of, say, Edward Hopper or Wayne Thiebaud with these techniques. We argue that economy is a good test of how well an algorithm can optimize strokes to match images well—a prerequisite for many natural painting styles.
>
> Hence, we view the problem of economical painting as foundational for many painting styles, because it’s really about developing general-purpose optimization techniques, and our method presents an advance on this problem. Applying them to more specific individual styles and providing better user controls are future work.
>
> A. In response to: “It would be interesting to see results on domain transfer, to see how specialized the learnt models are, including baselines”.
>
> Since we did not focus on domain adaptation or domain transfer experiments, we don’t expect our models to achieve great results on such tasks. Our current setup would require a further fine-tuning step to adapt the model to other domains. However, we consider this as future work.
>
> B. In response to: “Establish and discuss what limits the fidelity of the precise reconstructions” and “The 'precise' reconstructions are not particularly precise – they are significantly coarser than the originals, most prominently for landscapes. It would be good to see an analysis of why this is the case – e.g. is it due to the expressivity of 300 strokes, or optimisation issues?”
>
> We choose landscape subjects for economical painting stylization and facial portraiture to test the performance of our method on reconstructions. The former is common and well suited for stylization and the latter is well suited for realism/reconstructions. There is a performance gap of the same model in precise reconstructions from the two different subjects, and we believe this is in part due to the variance in the training images. While the CelebA dataset has a consistent distribution of high and low frequency parts in images, it is not the case in landscapes.
>
> The current network design presents a bottleneck between the encoder and decoder (512-d encoded vector to a sequence of Tx300 vectors, see Figure 3 (b-ii), that might be limiting the expressivity of the network to capture with high precision the amount of varying detail in landscapes, and further work would be needed to avoid such bottleneck and to achieve realistic paintings for this type of subject.
>
>
> C. In response to a direct optimization experiment request.
>
> It is non-trivial to achieve good results using optimization methods. We added a section in the appendix to show our findings. Randomizing a set of strokes and optimizing them to achieve a painting of a given input image does not work well. For instance, Zou et al. [1] use a common patching strategy with no more than 20 strokes per patch to be optimized. We have taken an alternative approach based on heuristics where strokes are initialized in a regular grid, and the start, middle and end control points of each stroke need to be relatively close to each other (short strokes) for this method to work (see Appendix section 9 in revised paper for more details). However, we believe that optimization methods without the use of neural networks are orthogonal to our line of work presented in this paper. While optimization methods do not need any training, they generally take minutes [2] to render a single painting.
>
> D. In response to the requested comparison with the Photoshop ‘oil paint’ filter.
>
> We appreciate the reviewer’s feedback and provide the requested comparison with the oil painting filter in Photoshop in Appendix, section 10.
>
> E.. In response to the requested changes with regards to writing and captions.
>
> We appreciate the reviewer’s feedback and we provide the requested changes in the new version of the paper. With regards to the caption in Figure 6, this is the only figure where we choose to show some images from the training set. All other figures show images from the test set.
>
> With regards to the organization of section 4, we follow the reviewer’s suggestions and place the training details section at the beginning of the section. We separate evaluation into an independent section. We also agree with the reviewer’s comment with regards to the naming of energy vs loss and use loss function in the paper.
>
> We address the other requested changes in the new paper.
>
> [1] Zou et al. Stylized neural painting.
>
> [2] Liu et al. Paint transformer

---

> > ### Comment · Reviewer_FXkV · 2022-07-28
> > **Response**
> >
> > I thank the authors for their detailed response, and the updated manuscript. These address a few of my concerns. However, the argument that economical painting is a-priori important (because certain artists' styles are economical) would be much more convincing if the paper displayed results that resembled one of the artists cited. I'm also concerned by Reviewer 6SHP's observation that the claims regarding being economical are overly strong, particularly in light of results with similar stroke count in Huang '19. As I previously noted, the technical contribution is also fairly small, and the quality of results leaves some room for improvement. Conversely, I found the paper generally well-written and enjoyable to read.

---

### Review · Reviewer_rhqr · 2022-06-28

**Summary Of Contributions:**

This paper tackles the problem of stroke-based painting synthesis from real images, with a particular focus on the "economical" aspect -- with a limited number of strokes. The main contribution is a new encoder-decoder framework that generates a sequence of strokes using either simple Fully connected layers or LSTMs, with various artistic styles ranging from precise depiction to abstraction controlled through different combinations of losses and optimization procedures.

The model seems to be able to produce compelling painting images with various styles using just Bézier curves.

**Broader Impact Concerns:**

I do not see immediate concerns on ethics or negative societal impacts.

**Requested Changes:**

## (Minor) Clarifications
- 2nd paragraph in Related Work: "Many recent methods employ neural optimization". Add some references.
- Below Eq (5), arc length $S_l=\| \mathbf{p}_1 - \mathbf{p}_2 \| + \| \mathbf{p}_2 + \mathbf{p}_3 \|$. Is it correct the second term is $\| \mathbf{p}_2 + \mathbf{p}_3 \|$? What does it mean?
- What are the datasets used for training?

**Strengths And Weaknesses:**

## Strengths
### Simple implementation
- The architecture is a simple encoder-decoder with either FC layers or LSTMs for as the decoder predicting the stroke sequences, and it is trained with only image loss without any painting image or stroke examples during training.
- This simple model seems to be able to produce pretty convincing painting results.

### Style variations
- The model can achieve different painting styles from precise depiction to abstract illustration by adjusting the loss formulations.

## Weaknesses
### Lacking principled way of controlling styles
- The paper describes different combinations of the losses with different weights and thresholds as well as different optimization strategies, for achieving various artistic styles. While it seems cool that the model can be tuned to achieve different styles, I still find it lacking some principled way of governing the styles, other than tuning specific parameters on the losses.

### Confusing figures
- Figures are poorly organized. Left and right of double column figures are organized in inconsistent structures. Multiple figures may also refer to the same thing in different parts, which makes the reference kind of messy, eg "Figure 4(c), Figure 6(left: second, third, and fourth) and Figure 7(c,g) show...". This is very confusing to read and makes it difficult to draw insights from the results.

### Biased evaluation on "economy"
- The paper seems to emphasize a lot on the "economical" aspect of the algorithm - ie, the number of strokes in a synthesized painting. It seems to me this is just one painting style, and I am a bit confused why this is used as an important evaluation metric to compare against other methods.
- The other metric that methods are compared on is the image reconstruction accuracy, which I also do not think is a conclusive metric for painting synthesis. Though I cannot think of a sensible and feasible evaluation protocol other than extensive qualitative evaluation.

### More brush types
- The model demonstrates only Bézier curve strokes. It would be interesting to see more results with other types of brushes as in PainTransformer.

---

> ### Author Response · Authors · 2022-07-12
> **Response to Reviewer rhqr**
>
> Common response to all reviewers:
>
> We want to emphasize that _economy_ is not just “one style.” Many common painting styles involve considerable precision in their brush strokes, e.g., consider the way the examples in Figure 1 use a few strokes that are carefully aligned to image features, curves follow contours, and so on. Existing painting algorithms are not economical; they often spray the canvas with strokes, using many strokes to reproduce the appearance of the image well, but often with an “impressionistic” look as a result; one could not imagine capturing the styles of, say, Edward Hopper or Wayne Thiebaud with these techniques. We argue that economy is a good test of how well an algorithm can optimize strokes to match images well—a prerequisite for many natural painting styles.
>
> Hence, we view the problem of economical painting as foundational for many painting styles, because it’s really about developing general-purpose optimization techniques, and our method presents an advance on this problem. Applying them to more specific individual styles and providing better user controls are future work.
>
> A. In response to: “...I still find it lacking some principled way of governing the styles, other than tuning specific parameters on the losses.”
>
> In our current setup, the style parameters have intuitive controls that result in a predictable effect. For instance, adding a small penalty on longer strokes results in smaller yet visible strokes in parts of the canvas that need more detail, but large strokes on background areas. Likewise, we expect that the addition of noise results in thinner and longer strokes where content is not represented accurately. However, we agree with the reviewer and more principled controls could be built on top of ours, or as a future work, for instance, learning from labeled styles or with presets.
>
> B. In response to having confusing figures:
>
> We appreciate the reviewer’s feedback. We have reorganized figure 5 and added letters to Figure 6 for better clarity.
>
> C. In response to having more brush types:
>
> We agree with the reviewer, but view this as future work and not necessary for this publication.
>
> D. In response to: “The paper seems to emphasize a lot on the ‘economical‘ aspect of the algorithm -ie, the number of strokes in a synthesized painting. It seems to me this is just one painting style.”
>
> Please, see the common response above. Economy is indeed one range of styles, one that we focus on because it is a proxy for how accurately an algorithm can optimize a few strokes well, rather than requiring many many small strokes in order to achieve good results. Previous methods require a large number of brush strokes scattered over an image, which doesn’t match many common painting styles, and being able to optimize economically is a prerequisite for capturing more styles in the future.
>
> E. In response to: “...I am a bit confused why this is used as an important evaluation metric to compare against other methods.” and “The other metric that methods are compared on is the image reconstruction accuracy, which I also don’t think is a conclusive metric for painting synthesis.”
>
> As the reviewer notes, there are no good evaluation metrics for image quality. The reviewer expresses appreciation for the results, which seems like as good a metric as any.
>
> As stated in the paper: “Simply comparing reconstruction error alone would provide a poor metric for painting algorithms”, we are not using reconstruction because it’s a good metric for image quality, it’s a measure of economy. As discussed in the preamble, our paper is an optimization paper, and reconstruction is a measure of optimization ability.
>
> F. In response to minor clarifications:
>
> 1. We have added references to the 2nd paragraph in related work, as suggested.
>
> 2. The reviewer noticed a typo in the equation below eq. 5, and we have modified the second term accordingly. The equation is then S_l = ||p1 - p2|| + ||p2 - p3||, where p1, p2 and p3 are the three control points of a quadratic bezier curve.
>
> 3. As stated in the last paragraph of section 4, training details: “We use 100000 landscapes training images gathered from Chen et al. (2018); Skorokhodov et al. (2021); Zhu et al. (2017) and 200000 CelebA Liu et al. (2015) training images.”

---

> > ### Comment · Reviewer_rhqr · 2022-07-24
> > **Response**
> >
> > I appreciate the authors' efforts in addressing my concerns, in particular explaining the importance of "economy" in painting synthesis. The paper focuses on a very specific aspect in "economy". Although I am no expert in painting synthesis, I still don't find the paper has clearly conveyed a compelling argument on why this is so important, and how the proposed method is significantly better. And as reviewer 6SHP pointed out, previous methods can also produce reasonable results with an limited budget of strokes.
> >
> > Overall, I'm still not fully convinced the contributions are significant enough and the evaluation is also not convincing enough at its current state.

---

### Review · Reviewer_6SHP · 2022-06-28

**Summary Of Contributions:**

The paper proposes a method for approximating images with “economical” paintings - that is, but a fairly small number of strokes (typically 300, in the context of this paper). The strokes are generated by an encoder-decoder model with a convolutional encoder and fully connected or LSTM decoder. The model is trained in supervised fashion with a combination of perceptual and other losses, using a differentiable renderer. The trained model does a good job at approximating images with the given stroke budget (similar or better than baseline approaches) and generates visually pleasing painting-like approximations in several styles achieved mostly by tweaks in the loss function.

**Broader Impact Concerns:**

No major ethical implications atm. Generally, on the upside, this research direction makes art more accessible and diverse; on the downside, it can lead to questions about fakes, in particular fake art. Moreover such models might eventually put some artists out of jobs.

**Requested Changes:**

Basically, to address the above concerns:
1. Polish the paper, make messaging clearer: explain why the focus on economical painting is justified, be even clearer about the main takeaway point(s)
2. Either provide additional quantitative results (user studies) on artistic qualities of the images, or revamp / tone down the corresponding claims made in the paper.
3. Make sure the constraints on the number of strokes is enforced on the baselines in a fair way. Report this in the paper.
4. Explain whether “stroke fine-tuning” is fair towards the baselines. If not, tone down the claims about SOTA performance of the proposed method

**Strengths And Weaknesses:**

Pros:
1. The paper is mostly well-written, the motivation and the method are quite clear
2. The method seems fairly simple, it does not depend on difficult-to-train techniques like reinforcement learning and adversarial training (although there are some experiments with a GAN loss in the paper)
3. The reconstruction quality is on par or above the baseline methods
4. There is some analysis of the importance of different components of the model

Cons/concerns:
1. I am unsure what exactly is the goal of the paper and what it teaches the reader.

a. My summary would be: good results on stroke-based reconstruction of images can be achieved without RL, but with simple supervised learning. If this is indeed the message,  I would encourage the authors to be more clear about it. It is mentioned among the key contributions, but it could be highlighted more clearly.

b. In particular, I am somewhat confused by the stress on economical painting. It seems prior methods experimented with similar stroke budgets of a few hundred strokes - how is the submission substantally different in this respect?

2. I am not entirely convinced by the experimental evaluation.

a. The only quantitative results are on reconstruction quality, while the paper also speaks a lot about artistic qualities of the reconstructions. If artistic qualities are indeed to be stressed, I would suggest user studies as an evaluation - unfortunately, eyeballing based on a few examples in the paper is not very reliable/robust. The paper even states “Our method focuses on economical-style paintings rather than aiming for a realistic version of a given input, common in previous painting methods.”, and then proceeds to use reconstruction quality as the main metric.

b. The baselines are constrained to 300 strokes same as the proposed method, for a fair comparison. How is this done? Are the baselines retrained with this number of strokes? If not, the comparison is not fair.

c. Without “stroke fune-tuning” the proposed method is not that great on landscapes (Table 2). Is this stroke fine-tuning fair compared to the baseline methods? In other words, coild one apply the same fine-tuning technique to the baselines and yield a similar improvement? If yes, comparing the baselines without fine-tuning to the proposed method with fine-tuning is not fair.

Smaller comments:
- It seems the usage of citet/citep (in-text/parenthetical citation) is inconsistent, for instance in the introduction many citations should be parenthetical
- I could not find the description of figure 1(i)?
- Weird sentence “If our method, based on a tight budget, can reconstruct the overall semantics and finer details of an input photograph, it should be, in principle, able to paint less strict forms of art.”
- Smth seems off with the caption of figure 8

---

> ### Author Response · Authors · 2022-07-12
> **Response to Reviewer 6SHP**
>
> Common response to all reviewers:
>
> We want to emphasize that _economy_ is not just “one style.” Many common painting styles involve considerable precision in their brush strokes, e.g., consider the way the examples in Figure 1 use a few strokes that are carefully aligned to image features, curves follow contours, and so on. Existing painting algorithms are not economical; they often spray the canvas with strokes, using many strokes to reproduce the appearance of the image well, but often with an “impressionistic” look as a result; one could not imagine capturing the styles of, say, Edward Hopper or Wayne Thiebaud with these techniques. We argue that economy is a good test of how well an algorithm can optimize strokes to match images well—a prerequisite for many natural painting styles.
>
> Hence, we view the problem of economical painting as foundational for many painting styles, because it’s really about developing general-purpose optimization techniques, and our method presents an advance on this problem. Applying them to more specific individual styles and providing better user controls are future work.
>
> A. In response to the request to justify the focus on economical painting and to make the message clearer.
>
> We agree with the reviewer and will provide a clarification of the goal of the paper. The goal of this paper is to provide an approach for economical painting styles, where strokes are generally more efficiently placed on the canvas. We believe this is more suitable for stylization than previous work where approximate an image via addition of many strokes. We aim for a simpler model which makes potential future work that builds on top of this work easier.
>
>
> B. In response to 1.b “...how is the submission substantially different from prior work?”
>
> Prior work does not focus on economical paintings. They use over a thousand strokes to achieve good reconstructions of an input image. Instead, we use 300 strokes and make the model architecture, loss functions and optimization process work specifically for economical paintings.
>
> C. In response to 2.a evaluation metric.
>
> As stated in the paper: “Simply comparing reconstruction error alone would provide a poor metric for painting algorithms”, we are not using reconstruction because it’s a good metric for image quality, it’s a measure of economy. As discussed in the preamble, our paper is an optimization paper, and reconstruction is a measure of optimization ability.
>
> D. In response to 2.b “... Are baselines retrained with this number of strokes”?
>
> We use three baselines:
>
> An optimization method that does not require training, Stylized Neural Painters (SNP). We limit the stroke number in the optimization procedure, thus ensuring the results are fair.
>
> A stroke prediction approach, Paint Transformer, which doesn’t train on real images nor uses a meaningful number of strokes during training. It doesn’t even use a related canvas resolution. Rather, this method generates a set of 8 random background strokes in a 32x32 canvas, and a second set of 8 foreground strokes. The network is tasked to correctly guess the foreground stroke parameters from the background set of strokes, and uses both a stroke level loss and a pixel loss. Since it trains on a random number of strokes, 8 in this case, it is not related to the final number of strokes at inference. At inference, the algorithm divides the input image into a set of patches, and proceeds to paint in a series of K scales. We adjust the algorithm so that the total number of strokes is 300, and thus, we ensure a fair comparison.
>
> A reinforcement learning approach. The baseline of this algorithm is trained with 200 strokes, and we retrain it to work with 300 strokes. At inference time, the algorithm divides the input image into a set of patches, and uses those patches as input to the network. Each output painting is a set of patches of 300 strokes each. We adjust the inference algorithm so that the total image has no more than 300 strokes. Thus, we ensure a fair comparison.
>
>
> E. In response to 2.c “... Is this fine-tuning fair compared to the baseline methods?”
>
> Stroke fine-tuning is added to landscapes paintings to achieve a higher level of resolution in the paintings, as an optional step. We decided to include this in table 2 because it is part of our method and to provide quantitative information about the fine-tuning step. However, the real comparison is before fine-tuning, and such a step is not included in previous methods. We adjust the bold letters in the revised version of the paper.
>
> F. In response to the smaller comments
>
> We appreciate the reviewers comments and we will provide the necessary modifications in the paper.
>
> E. In response to requested changes.
>
> We have rewritten some parts in introduction, evaluation, and conclusion, besides minor changes across the paper, following the suggestions of the reviewer.

---

> > ### Comment · Reviewer_6SHP · 2022-07-20
> > **thanks and updates to the review**
> >
> > I thank the authors for addressing my comments, in particular for clarifying the details on stroke budgets for baseline comparisons (seems fair) and stroke fine-tuning.
> >
> > However, some of my concerns remain:
> >
> > 1. The authors claim that "Prior work does not focus on economical paintings. They use over a thousand strokes to achieve good reconstructions of an input image.". This simply does not seem to be true: both l2p (RL-based) and SNP (optimization-based) experiment with - and get reasonable results - with budgets of 200-400 strokes, and in some cases substantially fewer too. The paper states ". Huang et al. (2019) can accurately reconstruct images, but requires thousands of tiny strokes to do so", but in the paper of Huang et al. there are clearly results with 200-400 strokes on CelebA/ImageNet
> >
> > 2. Given that stroke fIne-tuning is an extra step that shouldn't be taken into account for baseline comparisons, the method is actually not beating the baselines convincingly: it's slightly better than l2p on faces, and noticeably worse on landscapes on L1 loss (and basically on par on perceptual). Yet, the paper makes claims like:
> > - "this produces higher-precision results than previous methods"
> > - "Our method approximates stylistic versions of the input with fewer but with a more efficient placement of strokes than previous methods, where they often scatter thousand of strokes to approximate an input image"
> > - "We see that our method, in general, places strokes more efficiently than previous methods, which, in contrast, spray thousands of strokes to reproduce the input image
> >
> > I appreciate the fact that the proposed method is simple and works quite well, but at the moment presentation is misleading and I think the paper would have to be reworked substantially to improve this - in particular, dramatically reduce the stress on economical stylization being a distinguishing feature of this work and remove the best-performance claims (as well as ideally add more systematic evaluations of stylistic properties of the results)

---

### Author Response · Authors · 2022-08-16
**Response to the Decision by Paper173 Action Editors**

Dear Action Editors and reviewers,

We appreciate the reviewers and action editor’s feedback. We would like to emphasize some points:

The evaluation criteria stated in the TMLR Submission Guidelines and Editorial Policies says: “Papers should be accepted if they meet the criteria, even if the contribution or significance of the work is modest”.

We believe we make a modest contribution, enough to keep pushing the research in the area of painting algorithms and the reviewers’ comments support this. The decision here seems to violate the spirit of the TMLR policies.

The other comments mentioned in the reviews seem like little details, and any required changes can easily be adjusted in the text. It’s technically true that, as reviewer 6SHP mentions, l2p and SNP can produce paintings with low budgets. However, we empirically show in Figure 6 (right) that our model produces much clearer portraits than previous work with the same 300 stroke budget (compare row (b) with row (c) and (e)). Reviewer 6SHP seems to also use inconsistent language. In their last response, reviewer 6SHP writes: “… the method is actually not beating the baselines convincingly: it’s slightly better than l2p on faces, and noticeably worse on landscapes on L1 loss (and basically on par on perceptual)”. Table 1 shows that our method is better than l2p on L1 loss by 0.002, and perceptual on 0.018 on faces. Table 2 shows that our method is worse than l2p on L1 loss by 0.007 and that our method is better on perceptual also by 0.007 on landscapes. It seems like the reviewer is using different adjectives for the same delta (noticeably worse on L1 and basically on par on perceptual). Our method shows a difference in visual quality, mainly in Figure 6 (right), and very similar (if not better) quality in Figure 8 (compare columns c with d, e and f).

In our last response to reviewer 6SHP, section D, we clarified some confusion brought up by this reviewer previously, with regards to the fairness in comparison with previous methods, and reviewer 6SHP corroborates the fairness in our study in their first sentence of their last comment: “I thank the authors for addressing my comments, in particular for clarifying the details on stroke budgets for baseline comparisons (seems fair) and stroke fine-tuning”.
However, the last comment by the Action Editor states: “…, and the lack of necessary experimental results with the most relevant baselines in a fair setup”. It is unclear what this refers to..

We do believe this paper has the necessary empirical evidence, qualitative support, and ablation studies (supplemental material), to at least have a modest contribution in the field. If it is a matter of over-claiming contributions, we are happy to edit the text and tone-down our claims.

---

> ### Comment · Action_Editors · 2022-08-16
> **Response**
>
> Thanks for the note.  The rejection is not violating the TMLR policies.  The paper is not rejected because the contribution is modest; instead, it is rejected because the claims are not supported by accurate, convincing and clear evidence, the 1st question that every TMLR paper should answer.
>
> The claims are not supported in three ways.  First, the submission claims (or assumes) that economical painting is a valuable problem.  Reviewers, in particular FXkV and rhqr, have suggested that the definition of the concept is unclear, and the motivation is lacking.  The submission makes two additional claims that previous methods require many more strokes and do not perform as well with the same number of strokes.  As pointed out by reviewer 6SHP, neither of the claims is accurate.
>
> Regarding your specific point, I think your argument is incorrect and misleading, and reviewer 6SHP's point is correct.  It is true that on landscapes (Table 2), your model is worse than l2p by 0.007 on L1 and better by 0.007 on perceptual.  However, these metrics have different scales.  Your method's L1 loss is 17% higher (0.048 vs. 0.041), and its perceptual is only 1% better (0.658 vs. 0.651).  Therefore, the description that it is "noticeably worse on landscapes on L1 loss (and basically on par on perceptual)" seems accurate.
>
> Regarding the metareview, "the lack of necessary experimental results with the most relevant baselines" means the lack of superior results to support these best-performance claims, not the missing of the experiments themselves.
>
> Finally, I'd like to clarify that because the required changes are major (even the writing changes don't seem like they can be "easily adjusted"), and TMLR does not allow major revisions, the only reasonable decision is to reject the submission.  The metareview has suggested that we are willing to consider a significantly revised version of the manuscript.

---

### Decision · Action_Editors · 2022-07-29

**Recommendation:** Reject

**Comment:**

All three expert reviewers have reviewed the submission and rebuttal and recommended rejection.  All reviewers appreciate the simplicity of the method and the presentation of the paper.  However, shared concerns include the lack of a clear definition of "economical painting", some potential overclaim, and the lack of necessary experimental results with the most relevant baselines in a fair setup.  The AE agrees.  Based on the reviews, the AE has decided to recommend rejecting the submission, but encouraging the authors to resubmit the manuscript after a major revision to address the shared concerns.